# Eulerian Neural Network Informed by Chemical Transport for Air Quality Forecasting

**Xukai Zhang[1], Shuliang Wang[1], Guangyin Jin[2], Ziqiang Yuan[1], Hanning Yuan[1]\*, Sijie Ruan[1]\***

[1]Beijing Institute of Technology, [2]Sapienza University of Rome
{xkzhang,slwang2011,ziqiangy,yhn6,sjruan}@bit.edu.cn
jinguangyin96@gmail.com

## Abstract

Air pollution remains one of the most critical environmental challenges globally, posing severe threats to public health, ecological sustainability, and climate governance. While existing physics-based and data-driven models have made progress in air quality forecasting, they often struggle to jointly capture the complex spatiotemporal dynamics and ensure spatial continuity of pollutant distributions. In this study, we introduce CTENet, a novel chemical transport deep learning model that embeds the Advection-Diffusion-Reaction equation into a Physics-Informed Neural Network (PINN) framework using an Eulerian representation to model the spatiotemporal evolution of pollutants. Extensive experiments on two real-world datasets demonstrate that CTENet consistently outperforms state-of-the-art (SOTA) baselines, achieving a remarkable RMSE improvement of 45.8% on the USA dataset and 21.0% on the China dataset.

## 1 Introduction

Air quality refers to the concentration of pollutantssuch as $PM_{2.5}$, $PM_{10}$, ozone ($O_3$), sulfur dioxide ($SO_2$), nitrogen oxides ($NO_x$), and carbon monoxide ($CO$), and their impact on human health, ecosystems, and the environment. Exposure to these pollution is closely associated with respiratory and cardiovascular diseases, as well as increased risks of premature mortality[1, 2, 3]. Accurate air quality prediction is crucial for mitigating health risks, guiding public health, shaping policies, and enhancing environmental monitoring in smart, sustainable cities.

In recent years, extensive research has explored various modeling approaches for air quality prediction. Traditional physics-based models solve partial differential equations (PDEs) to simulate pollutant transport driven by atmospheric dynamics [4, 5]. While these models offer strong interpretability and theoretical grounding, they are often computationally expensive and sensitive to input accuracy. In contrast, simple data-driven models learn patterns directly from historical pollutant and meteorological observations [6, 7], but they may suffer from limited generalization in unseen or changing environments. Recent deep learning methods have further advanced this direction by modeling air quality as a multivariate time series at discrete stations, or by constructing graph representations of station networks to capture spatiotemporal dependencies [8, 9]. To improve predictive performance, these models typically incorporate auxiliary meteorological variablessuch as temperature and humiditycollected from ground-based monitoring sites. In addition, some multimodal approaches attempt to incorporate complementary information from heterogeneous sources to improve task understanding, robustness, and generalization [10, 11]. Beyond these data-driven models, physics-guided deep learning aims to combine the strengths of both approaches by incorporating physical knowledge into data-driven architectures[12, 13]. This hybrid paradigm seeks to enhance model accuracy and generalizability while ensuring physically plausible predictions.

---

\*Sijie Ruan and Hanning Yuan are the corresponding authors of this paper.

39th Conference on Neural Information Processing Systems (NeurIPS 2025).

Despite considerable progress, several core challenges in air quality forecasting persist:

**Structural mismatch between continuous Eulerian dynamics and discrete pollutant representations:** Whether or not they incorporate physical knowledge, most deep learning methods for air quality prediction adopt station-based modeling paradigms, where data are represented as multivariate time series or discrete graphs[14, 15, 16, 17, 18], as illustrated in Figure 1a. While these approaches effectively capture temporal dependencies and localized station interactions, they neglect the spatial continuity modeling of pollutant distributions. This limits their capacity to model the smooth and dynamic spatiotemporal evolution of air pollution over larger regions, shown in Figure 1b.

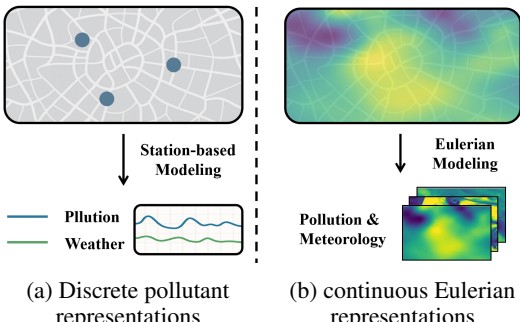

(a) Discrete pollutant representations

(b) continuous Eulerian representations

Figure 1: Two approaches to air pollution representation.

Additionally, the physical processes of diffusion and advection are inherently continuous and occur over continuous Eulerian space, rather than being naturally represented on discrete station graphs. Modeling these processes on graphs leads to a structural mismatch.

**Incomplete representation of physicochemical drivers:** Another critical limitation lies in the insufficient modeling of atmospheric chemical transformation processes and their meteorological drivers. While physical transport phenomena like advection and diffusion have received some attention in hybrid models[13, 19], the incorporation of complex atmospheric chemistrysuch as photochemical reactions and gas-to-particle conversion[20] that generate secondary pollutants (e.g., ozone and $PM_{2.5}$)remains limited , which are strongly modulated by meteorological factors such as solar radiation , boundary layer height, vertical temperature gradients. These variables are crucial for accurately simulating chemical reactions and phase transitions in secondary pollutant formation.

Unlike prior physics-guided models that rely on station-based or graph-based data, this paper proposes a novel approach performing Eulerian representation of pollutant evolution, enabling effective modeling of large-scale spatiotemporal diffusion, advection, and chemical transformations. It also integrates continuous meteorological data, including satellite-derived inversion, to enrich the modeling of underlying physicochemical dynamics.

Specifically, we propose the Chemical Transport Eulerian Network (CTENet), a deep learning architecture guided by chemical transport modeling (CTM) principles. CTENet embeds the Advection-Diffusion-Reaction (ADR) equation within a custom-designed neural network that operates on an Eulerian spatial domain, enabling direct simulation of pollutant transport and transformation over continuous space and time. The framework consists of several key modules: (1) the Eulerian Pollution Encoder, which encodes pollutant concentration fields within the continuous spatial domain; (2) the Meteorological Encoder, responsible for integrating high-dimensional meteorological data, including wind vector fields; (3) the Eulerian ADR Decoder, which decodes pollutant evolution through a physics-constrained differential equation network reflecting the ADR processes. In addition, we employ a modular and replaceable Spatiotemporal Sequence Predictor that functions as a Wind Predictor, Meteorology Predictor, and Pollutant Predictor within the overall framework.

In summary, the main contributions of this paper are as follows:

- We introduce a novel neural network, CTENet, which integrates a spatiotemporal sequence prediction network with a custom differential equation module, effectively capturing spatiotemporal variations in air quality. To the best of our knowledge, this is the first PINN approach for air quality prediction that adopts a spatial continuity perspective under the Eulerian framework.

- We integrate CTM with deep learning under an Eulerian framework to improve the modeling of spatiotemporal continuity in air quality data, explicitly simulating diffusion, advection, and reaction over continuous spatial domains for enhanced interpretability.

- We evaluate CTENet on real-world air quality datasets from China and the USA, achieving about 45.8% and 21.0% lower RMSE respectively compared to leading baselines.

## 2  Related Work

**Air Quality Prediction.**  Air quality prediction is fundamental for smart city development and urban planning, attracting extensive research efforts. Existing methods generally fall into two categories: physics-based models and data-driven models. The former category characterizes air pollution dynamicssuch as diffusion, advection, and chemical transformationthrough differential equations solved by numerical methods like finite difference or finite element methods [4, 5]. While these models offer interpretability grounded in physical laws, they often require substantial computational resources and depend on high-quality input data. Data-driven models leverage historical pollutant data and contextual information, including meteorological variables, to learn spatiotemporal dependencies. Early machine learning methods such as random forests and support vector machines showed initial success [21] but were limited in capturing complex spatiotemporal patterns. With advances in deep learning, architectures combining convolutional and recurrent networks (e.g., CNN-LSTM) [22] and graph-based networks [15, 16] have significantly improved prediction accuracy by modeling spatial and temporal dependencies more effectively. Attention-based models [17] and recent methods like Airformer [18] further enhance large-scale spatial correlation modeling, but they do not explicitly model the continuous spatial distribution of pollutants. In addition, some studies, such as STFNN [23] and AirRadar [24], emphasize modeling the spatial continuity of atmospheric processes, exhibiting strong capability in reconstructing and reasoning about pollutant fields, although they focus on inferring pollutant distributions rather than forecasting.

**Physics Guided Deep Learning.**  Recent advancements have explored the integration of physical principles into deep learning architectures [37, 38]. A prevalent approach involves incorporating physical knowledge into the loss function, wherein deviations from known physical laws are penalized to promote physical consistency and enhance model generalizability [39, 40]. However, the effectiveness of this method critically depends on the accuracy of the underlying physical knowledge. Inaccuracies can introduce inductive biases, thereby hindering the models representational capacity. Alternatively, hybrid frameworks have been proposed that embed scientific knowledge into specific components of the model architecture, enabling a more flexible and potentially more robust integration of physics into the learning process [41, 42]. In recent air quality forecasting studies, physics-guided frameworks have been increasingly adopted to better capture the underlying dynamics of pollutant transport and transformation. For example, some studies have incorporated advection-diffusion mechanisms and fluid dynamics constraints into learning models [12]. Other works introduced state-space concepts inspired by physical systems to represent temporal evolution [43]. Meanwhile, several approaches [13, 19] have modeled air quality dynamics as graph-based systems constrained by advectiondiffusion processes, enhancing interpretability and generalization. However, the physical processes are limited to the graph edges, without modeling spatial continuity.

## 3  Problem Formulation

Given historical pollutant concentration data $P_{1:T} \in \mathbb{R}^{T \times C_P \times N}$ from $N$ observation stations located at spatial coordinates $\mathcal{S} = \{(h_n, w_n)\}_{n=1}^{N}$, and the corresponding continuous meteorological data $M_{1:T} \in \mathbb{R}^{T \times C_M \times H \times W}$, where $C_M$ denotes the number of channels, including those for east-west and north-south wind components as well as other meteorological variables.

We aim to predict the pollutant concentrations at all station locations over the future time period from $T + 1$ to $T + \tau$, denoted by $\hat{P}_{T+1:T+\tau}$. Formally, the task is to learn the function $\mathcal{F}$ :

$$\hat{P}_{T+1:T+\tau} = \mathcal{F}(P_{1:T}, M_{1:T}), \quad \hat{P}_{T+1:T+\tau} \in \mathbb{R}^{\tau \times C_P \times N} \tag{1}$$

## 4  Methodology

### 4.1  Model Overview

In this paper, we propose the Chemical Transport Eulerian Network (CTENet), as illustrated in Figure 2. The framework comprises three key components: the Eulerian Pollution Encoder, the Meteorological Encoder, and the Eulerian ADR Decoder. Additionally, a replaceable Spatiotemporal Sequence Predictor functions as the Wind, Meteorology, and Pollutant Predictor in the framework.

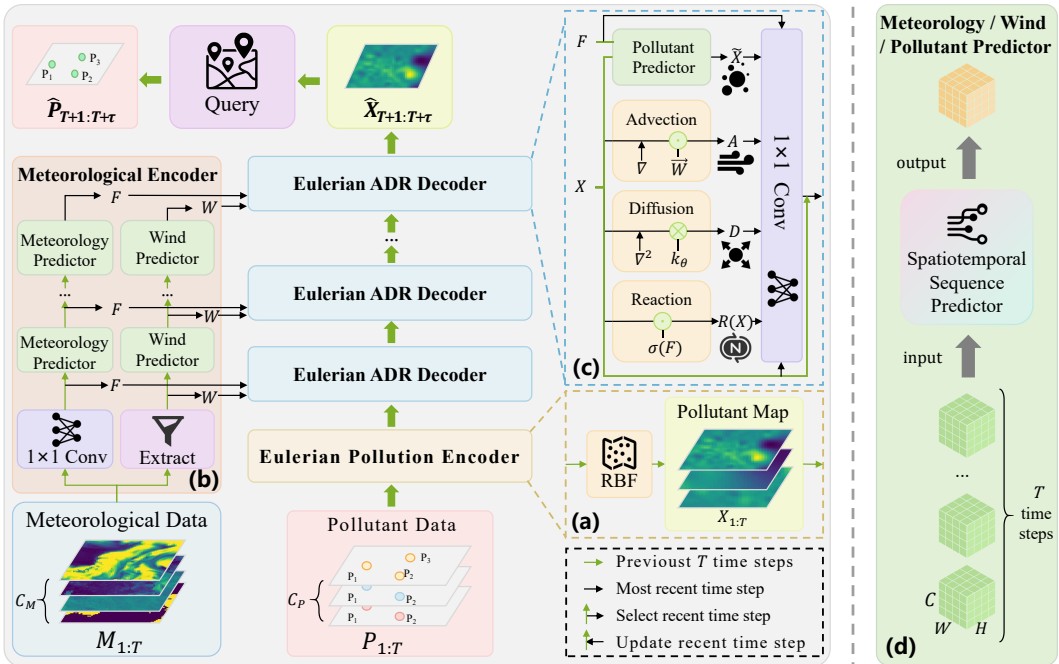

Figure 2: Structure of CTENet. Pollutant data is processed by the (a) Eulerian Pollution Encoder, while meteorological data are fed into the (b) Meteorological Encoder. The outputs are jointly passed to the multi-layer (c) Eulerian ADR Decoder, followed by a query step to generate the final output. (d) Spatiotemporal Sequence Predictor represents the general building block used in the three forecasting modules.

## 4.2 Eulerian Pollution Encoder

At each time step, pollutant concentrations from monitoring stations are assigned to their spatial locations, averaging values when multiple stations fall within the same pixel. However, these monitoring stations are sparsely distributed and cannot fully cover the entire spatial domain, as shown in Figure 3. For locations without monitoring stations, we perform Radial Basis Function (RBF) interpolation to generate a continuous Eulerian representation of pollutant distribution $X$, as shown in Figure 4. This continuous representation facilitates the modeling of spatial pollutant dynamics and serves as a proxy for learning air quality prediction over the entire domain.

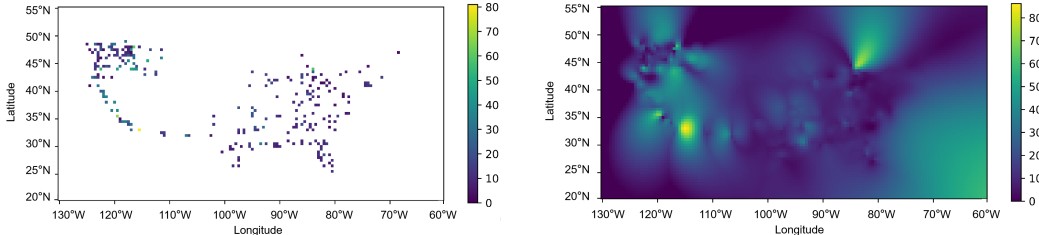

Figure 3: A sample of ground truth for $PM_{2.5}$ concentrations across the continental United States

Figure 4: A sample of RBF interpolation result for $PM_{2.5}$ concentrations across the continental United States

Compared to other interpolation methods, RBF is particularly well-suited for modeling air pollutant data due to its flexibility, smoothness, and ability to handle irregularly distributed monitoring stations. It provides a kernel-based and distance-based interpolation framework that adapts well to the complex, non-linear spatial variability often observed in environmental data, making it ideal for constructing continuous inputs for deep learning models.

The fundamental idea of RBF interpolation is to infer a continuous field from discrete observations by expressing the spatial relationships among data points through radial basis functions. Specifically,

the interpolated value at a spatial location $\mathbf{x}$ is computed as:

$$rbf(\mathbf{x}) = \sum_{i=1}^{n_t} \lambda_i^{(t)} \phi\left(\left\|\mathbf{x} - \mathbf{x}_i^{(t)}\right\|\right), \quad t \in \{1, \ldots, T\} \tag{2}$$

Here, $\mathbf{x}_i$ represents the position of the $i$-th known data point, and $\|\mathbf{x} - \mathbf{x}_i^{(t)}\|$ denotes the Euclidean distance between $\mathbf{x}$ and $\mathbf{x}_i^{(t)}$. The function $\phi(\cdot)$ is the chosen radial basis function, and $\lambda_i^{(t)}$ is the interpolation weight for each data point. To determine the weights $\lambda^{(t)} = [\lambda_1^{(t)}, \ldots, \lambda_n^{(t)}]^\top$, we solve the following linear system based on the known observations $y_i^{(t)}$ at the locations $\mathbf{x}_i^{(t)}$:

$$\Phi^{(t)} \lambda^{(t)} = y^{(t)}, \quad t \in \{1, \ldots, T\} \tag{3}$$

where $\Phi_{ij}^{(t)} = \phi\left(\|\mathbf{x}_i^{(t)} - \mathbf{x}_j^{(t)}\|\right)$ is the kernel matrix constructed from pairwise distances between known data points, and $y^{(t)} = [y_1^{(t)}, \ldots, y_n^{(t)}]^\top$ is the vector of known pollutant values.

In this paper, we adopt the **Multiquadric** function as the radial basis function, defined as:

$$\phi(r) = \sqrt{r^2 + c^2} \tag{4}$$

where $r = \|\mathbf{x} - \mathbf{x}_i^{(t)}\|$ and $c > 0$ is a smoothing parameter that controls the flatness of the basis function. The Multiquadric kernel is particularly effective in capturing smooth spatial patterns in complex environmental fields.

## 4.3  Meteorological Encoder

Meteorological data $M$ is processed by extracting wind features for ADR modeling and applying channel-wise transformation to allow interactions among heterogeneous meteorological variables, yielding a compact yet informative feature representation. Below, we detail these two processing methods.

First, we extract the wind-related channels from $M$, denoted as $\vec{W} \in \mathbb{R}^{T \times 2 \times H \times W}$, where the two channels represent the eastwest and northsouth components of horizontal wind. These wind features are essential for capturing directional wind patterns that influence pollutant dispersion.

In parallel, we apply a $1 \times 1$ convolution over the full meteorological tensor $M$, acting on the channel dimension, to reduce the number of channels while enhancing the representational capacity:

$$F = Conv_{1\times1}(M), \quad F \in \mathbb{R}^{T \times C_F \times H \times W} \tag{5}$$

where $C_F$ denotes the number of output channels in the transformed meteorological representation. Since future weather conditions have a significant impact on air quality, meteorological forecasting is integrated into the model. The *Meteorological Encoder* includes two specialized components: the *Wind Predictor*, which forecasts wind vectors $\vec{W}$, and the *Meteorology Predictor*, which predicts general meteorological features $F$. Each predictor can adopt any **Spatiotemporal Sequence Predictor**, such as ConvLSTM [44] or TAU [45]. The inputs to these predictors are four-dimensional tensors of shape $(T, C, H, W)$ as shown in Figure 2 (d), where $C$ is the number of channels specific to each predictor depending on the input variable types. Each predictor generates a one-step-ahead forecast, producing an output with a time dimension of 1. The predicted outputs from both components are subsequently incorporated into the *Eulerian ADR Decoder* to enhance predictive performance and ensure better alignment between future meteorological conditions and pollutant dispersion patterns.

## 4.4  Eulerian ADR Decoder

Eulerian ADR Decoder integrates crucial domain knowledge from atmospheric sciences, particularly Chemical Transport Models (CTM)[46], to simulate pollutant dynamics grounded in physical and chemical principles. Central to CTM is the AdvectionDiffusionReaction (ADR) equation[47], which governs the spatiotemporal evolution of pollutants in the atmosphere.

The main idea of ADR equation is to describe how pollutant concentrations change over time and space due to four fundamental mechanisms: advection, diffusion, chemical reaction, and external

source term such as sources and sinks. The continuous form of ADR equation can be expressed as:

$$\frac{\partial X}{\partial t} + \underbrace{\vec{W} \cdot \nabla X}_{Advection} = \underbrace{k_\theta \cdot \nabla^2 X}_{Diffusion} + \underbrace{R(X)}_{Reaction} + \underbrace{S}_{Source} \tag{6}$$

where $k_\theta$ is the diffusion coefficient. The mathematical analysis of ADR equation can be found in Appendix H,and the boundary conditions in I .

To enable numerical implementation, we discretize the continuous ADR equation 6 using the implicit forward-time central-space (FTCS) finite difference method [48] to approximate the spatial and temporal derivatives.

**Spatial discretization:** We apply the central difference method [49], which allows us to approximate spatial gradients and diffusion terms in a numerically stable manner:

$$\nabla X \approx \left( \frac{X[i+1,j] - X[i-1,j]}{2\Delta x}, \frac{X[i,j+1] - X[i,j-1]}{2\Delta y} \right) \tag{7}$$

$$\nabla^2 X[i,j] \approx \frac{X[i+1,j] + X[i-1,j] + X[i,j+1] + X[i,j-1] - 4X[i,j]}{4\Delta x \Delta y} \tag{8}$$

Here, $X[i,j]$ refers to the pollutant concentration at the grid point $(i,j)$, and $\Delta x$, $\Delta y$ are the spatial resolutions in the longitude and latitude directions, respectively.

**Temporal discretization:** We adopt the explicit Euler method [49] for temporal discretization. The explicit Euler update for the pollutant concentration $X$ is given by:

$$\hat{X}_{T+1} = X_T + \Delta t \left( -\vec{W}_T \cdot \nabla X_T + k_\theta \cdot \nabla^2 X_T + R(X_T) + S_T \right) \tag{9}$$

where $\Delta t$ is the time step size, $X_T$ is the concentration at time step $T$, and $X_{T+1}$ is the updated concentration at the next time step.

For different terms, we adopt explicit and implicit modeling strategies based on real-world considerations. The following introduces each term in equation 9:

**Advection term** is represented as $A_{T+1} = -\vec{W}_T \cdot \nabla X_T$, modeling the transport of pollutants due to the wind. The advection term accounts for the movement of pollutants along the direction of the wind, simulating the large-scale, directional transport across the spatial domain. Explicitly modeling advection allows the system to capture the influence of wind fields on pollutant movement, making it especially crucial for accurate spatiotemporal predictions in areas where wind plays a significant role in pollutant dispersion. This is particularly important in scenarios involving strong wind fields or long-range pollutant transport, where pollutants can travel vast distances within short periods.

**Diffusion term** is represented as $D_{T+1} = k_\theta \cdot \nabla^2 X_T$, where the diffusion coefficient $k_\theta$ is learnable here to capture dynamic diffusion behavior during training. This term models the random spread of pollutants due to molecular motion or turbulence, accounting for the gradual diffusion of pollutants across areas with varying concentration levels. The diffusion term is essential for capturing local mixing effects and pollutant dispersion in regions with weak wind or when pollutants encounter obstacles that hinder direct transport. By incorporating a learnable diffusion coefficient, the model can adapt to different environmental conditions and capture the dynamic nature of pollutant dispersion.

**Reaction term** models nonlinear chemical transformations among pollutants, such as oxidation reactions, photochemical reactions, and secondary particulate matter formation. These reactions are highly sensitive to atmospheric conditions, including temperature, solar radiation, and humidity, among others. Instead of attempting to model thousands of complex chemical reaction equations, which are difficult to construct accurately, we opt for a more flexible approach. Meteorological features $F_{1:T}$ are mapped through a sigmoid activation, yielding modulation coefficients constrained to the range $(0, 1)$, which can be interpreted as a form of *environment-guided soft attention* applied along the channel dimension. Specifically, the chemical reaction term is computed by element-wise multiplying the pollutant concentrations with the corresponding modulation coefficients, as shown below:

$$R(X_{T+1}) = \sigma(F_T) \odot X_T \tag{10}$$

**Source term** is represented as $S_{T+1}$,which is not explicitly parameterized in our model, but implicitly captured by the deep neural network. This allows the model to flexibly learn complex spatiotemporal emission patternssuch as anthropogenic and biogenic sources or depositionbased on

observational data and meteorological inputs, without relying on fixed or analytical formulations of typically inaccessible or sparsely available factors.

Additionally, an intermediate representation $\tilde{X}_{T+1}$ is generated by applying the *Pollutant Predictor* to the historical sequence $X_{1:T}$. The *Pollutant Predictor* shares a similar architecture with the *Wind Predictor* and the *Meteorology Predictor*.

$$\tilde{X}_{T+1} = PollutantPredictor\left(X_{1:T} \parallel F_{1:T}\right), \quad \tilde{X}_{T+1} \in \mathbb{R}^{1 \times C_P W \times H \times W} \tag{11}$$

The vertical bar $\parallel$ denotes concatenation along the channel dimension.

In contrast to the additive formulation in equation 9, we employ a modified approach where the components are concatenated along the channel dimension, allowing the model to flexibly learn complex, nonlinear interactions between features while preserving their individual information before fusion. In addition, we concatenate the predicted future state $\tilde{X}_{T+1}$, which serves as a proxy for the true future state, with the meteorological features $F_T$, providing the model with a richer context that goes beyond the traditional ADR modeling capabilities. Finally, a $1 \times 1$ convolution is applied to generate the final prediction $\hat{X}_{T+1}$:

$$\hat{X}_{T+1} = Conv_{1\times1}\left(X_T \parallel \Delta t\left(A_{T+1} \| D_{T+1} \| R(X_{T+1})\right) \parallel \tilde{X}_{T+1} \parallel F_T\right) \tag{12}$$

### 4.5 Query and Optimization

Upon obtaining the predicted Eulerian representation $\hat{X}_{T+1} \in \mathbb{R}^{1 \times C_P \times H \times W}$, we query the values at the station locations to obtain the final output as:

$$\hat{P}_{T+1} = Query(\hat{X}_{T+1}, \mathcal{S}), \quad \hat{P}_{T+1} \in \mathbb{R}^{1 \times C_P \times N} \tag{13}$$

CTENet performs recursive multi-step forecasting of pollutant concentrations over a future horizon. At each iteration, it takes the past $T$ time steps of input to predict the pollutant concentration $\hat{X}$, and uses a sliding window mechanism to advance the input sequence by one step—replacing the oldest input with the latest prediction. By applying this process iteratively, CTENet generates the multi-step prediction sequence $\hat{P}_{T+1:T+\tau} \in \mathbb{R}^{\tau \times C_P \times N}$.

For optimization, we minimize the discrepancy between the predicted and ground-truth pollutant concentrations at the station locations over the entire forecast horizon. Specifically, the model, parameterized by $\Theta$, is trained by minimizing RMSE loss:

$$\mathcal{L}(\Theta) = \sqrt{\frac{1}{\tau N} \sum_{t=T+1}^{T+\tau} \sum_{n=1}^{N} \left\| P_t^{(n)} - \hat{P}_t^{(n)} \right\|_2^2} \tag{14}$$

where $P_t^{(n)}$ and $\hat{P}_t^{(n)}$ denote the ground truth and predicted pollutant concentration vectors at station $n$ and time $t$.

## 5 Experiments

We compare our proposed CTENet with the representative and SOTA models to evaluate their effectiveness on two real-world pollutant datasets. Following the common practice in previous related studies, this study focuses on $PM_{2.5}$ concentration as the target variable, with meteorological data serving as auxiliary variables. Unless otherwise specified, TAU [45] is used as the default *Spatiotemporal Sequence Predictor* type in all experiments. Further experiments details are provided in the Appendix C and D.

**Dataset:** The pollutant monitoring station datasets are collected from China[18] and the United States[50], covering the entire year of 2018. Simultaneously, we obtain meteorological data from the National Centers for Environmental Prediction (NCEP) [51] for the same time and locations.

**Baselines:** We choose the baseline methods from three categories: (1) Statistical models, including Historical Average (HA) and Vector Auto-Regression (VAR)[52]. (2) Graph-based models, including STGCN [53], DCRNN [54], GTS [55], AirFormer [18], AirPhyNet [13], and $PM_{2.5}$-GNN [16].

(3) Spatiotemporal sequence prediction models, including TAU. M To ensure fair comparisobn with site-based models, we assign the pixel value at each grid cell to all stations located within that cell. More details can be found in Appendix C.

**Implement Details:** Each dataset is divided into training, validation, and test sets in a $7 : 1.5 : 1.5$ ratio. We use data from the past 72 hours to predict pollutant concentrations for the next 24, 48, and 72 hours. Our implementation code is released for public use[2].

## 5.1 Main Results

Table 1 presents a performance comparison between CTENet and several baseline methods across two datasets. We can draw the following observations: (1) CTENet outperforms the other baseline models in terms of overall performance, regardless of whether ConvLSTM or TAU is used as the prediction module. Specifically, compared to the best-performing baseline, our best CTENet variant achieves an average MAE improvement of 33.1% and an RMSE improvement of 45.8% in the USA, while in China, it achieves an average MAE improvement of 15.8% and an RMSE improvement of 21.0%. This substantial improvement underscores the effectiveness and superiority of the CTENet framework. Notably, when using TAU as the predictor, CTENet achieves optimal results in most cases; (2) Graph-based deep learning-based methods clearly outperform traditional statistical methods in prediction accuracy, demonstrating the powerful capability of deep learning in learning complex spatiotemporal relationships and modeling nonlinear patterns; (3) Spatio-temporal deep learning models originally developed for traffic forecasting, such as GTS and STGCN, exhibit suboptimal performance in our experiments. This may be attributed to their lack of specialized architectural adaptations for processing high-dimensional meteorological inputs. (4) Spatiotemporal sequence prediction models, such as TAU, show great potential in fine-grained air quality prediction tasks, performing on par with SOTA graph-based models.

Table 1: $PM_{2.5}$ prediction performance: the best-performing result is highlighted in bold, while the second-best is underlined for easy comparison.

| Methods | USA Data | | | | | | China Data | | | | | |
| --- | --- | --- | --- | --- | --- | --- | --- | --- | --- | --- | --- | --- |
| | 24h | | 48h | | 72h | | 24h | | 48h | | 72h | |
| | MAE | RMSE | MAE | RMSE | MAE | RMSE | MAE | RMSE | MAE | RMSE | MAE | RMSE |
| HA | 5.30 | 11.57 | 5.66 | 12.54 | 5.99 | 13.23 | 21.64 | 38.03 | 22.76 | 39.12 | 23.58 | 40.03 |
| VAR | 6.32 | 14.41 | 5.78 | 12.74 | 5.76 | 12.94 | 24.74 | 39.85 | 25.43 | 41.85 | 26.66 | 44.14 |
| STGCN | 4.29 | 9.03 | 4.51 | 9.03 | 4.63 | 9.08 | 31.43 | 43.72 | 31.91 | 44.06 | 32.69 | 44.75 |
| DCRNN | 5.40 | 14.50 | 5.42 | 12.81 | 5.38 | 13.48 | 28.14 | 49.81 | 27.45 | 47.36 | 27.39 | 47.63 |
| GTS | 5.57 | 14.65 | 5.60 | 14.32 | 5.61 | 14.18 | 23.46 | 41.70 | 23.50 | 42.53 | 23.85 | 44.41 |
| AirFormer | 4.05 | 10.44 | 4.40 | 10.74 | 4.60 | 10.89 | 19.09 | 36.08 | 20.89 | 38.42 | 21.85 | 39.61 |
| AirPhyNet | 4.47 | 11.36 | 4.79 | 11.40 | 4.94 | 11.48 | 18.75 | 36.35 | 19.97 | 37.16 | 20.74 | 37.64 |
| $PM_{2.5}$-GNN | 4.38 | 9.77 | 4.63 | 9.66 | 4.76 | 9.63 | 17.71 | 33.25 | 19.12 | 34.16 | 19.73 | 34.53 |
| TAU | 4.71 | 12.51 | 4.94 | 13.56 | 5.22 | 13.90 | 15.85 | 26.80 | 15.43 | 27.35 | 15.60 | 26.85 |
| CTENet w/ ConvLSTM | 4.12 | 8.46 | 4.31 | 8.66 | 4.43 | 8.84 | 13.79 | 23.14 | 14.44 | 23.79 | **15.28** | **24.47** |
| CTENet w/ TAU | **2.66** | **4.86** | **2.99** | **4.86** | **3.10** | **5.00** | **10.90** | **16.99** | **13.28** | **22.60** | 15.92 | 26.74 |
| % Best Improvement | 34.43 | 46.02 | 32.06 | 46.18 | 32.68 | 44.86 | 31.24 | 36.60 | 13.97 | 17.36 | 2.04 | 8.86 |

Overall, these experimental results validate the exceptional performance of CTENet in air quality prediction tasks. By combining a CTM-based pollutant model with an Eulerian spatiotemporal modeling framework, CTENet effectively leverages spatial continuity and temporal dynamics, enabling more accurate and physically consistent air quality predictions.

## 5.2 Ablation Study

To evaluate the individual contributions of each component in the model for 24-hour predictions, we conduct a systematic ablation study by progressively removing components. Figure 5 presents the ablation results for the ADR terms, while ablation experiments related to the interpolation method are provided in Appendix F.

**Effect of Advection and Difussion term:** Full CTENet outperforms models with the removal of the diffusion or advection modules on both the China and USA datasets, highlighting the benefits of incorporating physical knowledge. The removal of the diffusion term leads to a relatively minor degradation in performance, particularly in the China dataset, possibly because the multilayer CNNs in the predictor are able to partially compensate for its absence by implicitly modeling local spatial dependencies. The theoretical proof is provided in Appendix B. In contrast, removing the Advection

---

[2]`https://github.com/santafirefox0/CTENet`

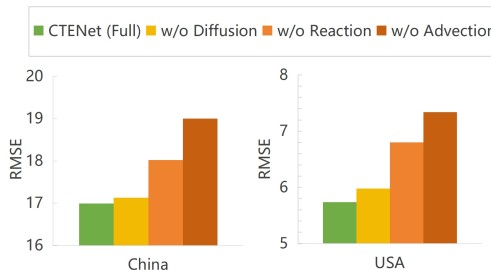

Figure 5: Ablation study of ADR terms.

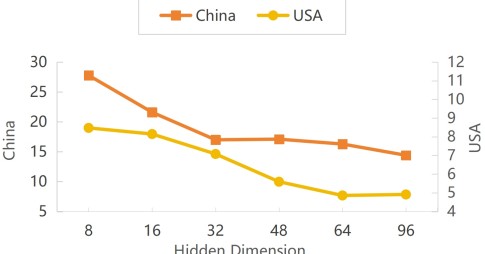

Figure 6: Hidden dimension analysis on RMSE.

module causes a more significant performance decline, emphasizing the difficulty standard deep networks face in learning pollutant transport due to wind, and the necessity of integrating domain-specific physical mechanisms into the model architecture.

**Effect of Reaction term:** Removing the reaction term results in a noticeable performance decline, even though historical meteorological features remain part of the input. This indicates that the benefit of reaction term lies not in simply incorporating additional features, but in its role as a dynamic weighting mechanism. By adaptively modulating the influence of meteorological inputs, the reaction term enhances the model's ability to represent complex pollutant dynamics, highlighting the value of structured chemical integration in CTENet.

### 5.3 Hidden Dimension Analysis

In this experiment, we evaluate the RMSE performance of CTENet under different hidden dimension settings for two datasets, as illustrated in Figure 6. The results show that increasing the hidden dimension generally leads to lower RMSE on both datasets, indicating improved model performance. However, the improvement is not indefinite: for the China dataset, the RMSE plateaus beyond a hidden dimension of 32, while the USA dataset achieves its lowest RMSE at 64. This suggests that increasing the hidden dimension yields diminishing returns, as model complexity eventually exceeds the intrinsic structure of the data, potentially leading to overfitting. Moreover, the optimal hidden dimension varies across datasets, likely due to differences in data complexity or noise characteristics.

### 5.4 Case Study

To demonstrate the interpretability and physical consistency of our model, we present a case study based on predicted $PM_{2.5}$ concentrations over central China. Figure 7 displays the spatial distribution of $PM_{2.5}$ at four consecutive time steps. In the highlighted red box, northwest winds are observed at the first three time steps. As a result, the pollutant mass shifts toward the southeast in the subsequent frame, indicating that the trans-

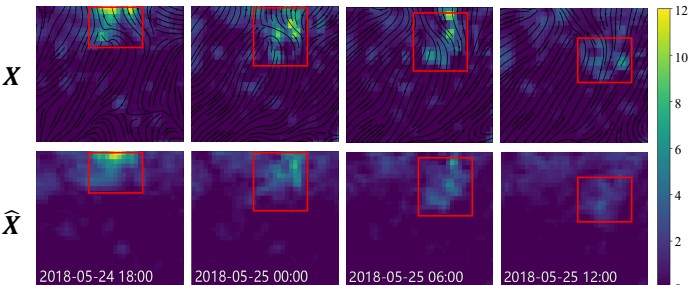

Figure 7: Visualization of $PM_{2.5}$ concentrations as heatmaps. The first row shows reference distributions $X$ overlaid with wind streamlines, and the second row shows model predictions $\hat{X}$.

port of $PM_{2.5}$ is consistent with the wind direction. This spatial displacement of pollution illustrates how CTENet captures the dynamic advection of pollutants, consistent with atmospheric physical principles, and reflects the model's capacity to learn meaningful spatiotemporal interactions.

## 6 Conclusion and Future Work

In this work, we propose the CTENet network for air quality prediction, which integrates physics-guided models with deep learning techniques to effectively model the spatiotemporal evolution of pollutants. Comprehensive experiments on several benchmark datasets have validated the effec-

tiveness of the proposed method, achieving a remarkable RMSE improvement of 45.8% on the USA dataset and 21.0% on the China dataset. Our current formulation lacks uncertainty quantification, limiting its applicability in risk-sensitive scenarios such as probabilistic forecasting or decision-making under uncertainty. Future work will explore integrating uncertainty-aware mechanisms to improve reliability and interpretability.

## Acknowledgments and Disclosure of Funding

This research is supported by National Natural Science Foundation of China (No. 62306033, 42371480).

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

## A  Notation

Table 2: Summary of notations used in the paper

| Symbol | Description |
|---|---|
| $N$ | Number of observation stations |
| $T, \tau$ | Input time step and prediction time step, respectively |
| $H, W$ | Total number of grid points in the latitude and longitude dimensions |
| $P_t$ | Historical pollutant concentration data at time $t$ |
| $X_t$ | Spatially interpolated historical pollutant concentration data at time $t$ |
| $M_t$ | Continuous historical meteorological data at time $t$ |
| $C_P$ | Number of pollutant types and the number of pollutant channels |
| $C_M$ | Number of meteorological channels |
| $C_F$ | Number of meteorological feature channels. |
| $\vec{W}_t$ | Continuous wind vector at time $t$ |
| $F_t$ | Continuous meteorological features at time $t$ |
| $S_t$ | Source Term at time $t$ |
| $R(\cdot)$ | Function modeling the chemical transformation of pollutants |
| $k_\theta$ | Learnable diffusion coefficient |
| $\odot$ | Element-wise product |
| $\sigma(\cdot)$ | Sigmoid function: $\sigma(x) = \frac{1}{1+e^{-x}}$, used to squash values between 0 and 1. |
| $\parallel$ | Channel-wise concatenation |

## B  Proof of Convolution Kernel Fitting Diffusion

The diffusion term $k_\theta \cdot \nabla^2 X$ is formulated as the product of the diffusion coefficient $k_\theta$ and the Laplacian operator of the pollutant concentration $X$, whose discrete form is defined in Equation (8). In this section, we demonstrate how a $3 \times 3$ convolution operation in a convolutional neural network can approximate the discretized Laplacian operator. The convolution operation can be viewed as a weighted summation of spatial data using a sliding window, which can be expressed as:

$$Y[i,j] = \sum_{m=-1}^{1} \sum_{n=-1}^{1} K[m,n] \cdot X[i+m, j+n] \tag{15}$$

where $K$ is the convolution kernel, $X[i,j]$ is the input data, and $Y[i,j]$ is the result of the convolution operation. The model ultimately learns an appropriate convolution kernel $K$, which, when close to the following form, causes the convolution operation to approximate the Laplacian operator:

$$K = \begin{bmatrix} 0 & 1 & 0 \\ 1 & -4 & 1 \\ 0 & 1 & 0 \end{bmatrix} \tag{16}$$

This completes the proof. A related empirical demonstration can be found in [56].

## C  Baselines Details

We compare our CTENet with SOTA models for long-term graph sequence prediction and spatiotemporal sequence prediction models, some of which are specifically optimized for air quality prediction.

- **HA** uses the average of the historical observations as the predictions. This simple baseline can provide insights into the effectiveness of more complex models.

- **VAR** [52] is a classical statistical model that was originally proposed for macroeconomic analysis. It utilizes past values of the multivariate time series to make predictions. It captures temporal dependencies without considering the spatial aspect.
- **STGCN** [53] integrates both spatial and temporal dependencies, leveraging graph-based convolutions to predict air quality over time.
- **DCRNN** [54] improves on STGCN by incorporating recurrent layers and diffusion-based graph convolutions to capture dynamic relationships in both spatial and temporal domains.
- **GTS** [55] proposes a method for simultaneously learning the graph structure and forecasting multivariate time series using GNNs, where the graph is unknown, by optimizing a probabilistic graph model.
- **Airformer** [18] is a novel Transformer-based model designed to predict nationwide air quality in China with fine spatial granularity, incorporating a two-stage learning process to efficiently capture spatio-temporal representations and intrinsic uncertainty.
- **AirPhyNet** [13] is a novel approach that integrates physical principles of air particle movement (diffusion and advection) into a neural network architecture on graph-based data structures, improving air quality prediction by capturing spatio-temporal relationships and providing interpretable, physically meaningful predictions.
- **$PM_{2.5}$-GNN** [16] is a novel graph-based model designed to capture both fine-grained and long-term dependencies in $PM_{2.5}$ forecasting, incorporating critical domain knowledge and validated on real-world data, with an online deployment for free forecasting service.
- **TAU** [45] decomposes temporal attention into intra-frame statical attention and inter-frame dynamical attention to efficiently capture both spatial dependencies within a single frame and temporal dependencies across frames, enabling parallelization of the temporal module in spatiotemporal predictive learning.After referring to [57, 58], we have also decided to adopt TAU as one of the predictors in our model.

The key hyperparameters in other baselines are also tuned for the best performance.

## D   Experimental Setup

**Datasets Details:** We conduct extensive experiments on two real-world pollutant monitoring station datasets collected from China and the United States, covering the entire year of 2018, to evaluate the performance of our proposed CTENet. The Chinese dataset[18] includes 480 primary monitoring stations. The United States dataset[50] consists of 365 monitoring stations across the contiguous United States. Both datasets provide hourly air quality. Simultaneously, we obtain meteorological data from the National Centers for Environmental Prediction (NCEP) [51] for the same time and locations, covering 354 channels with information on temperature, humidity, wind speed, potential height, vertical velocity, sunlight, cloud cover, and other variables at different isobaric levels, with a spatial resolution of 0.5ř in both latitude and longitude.

**Implementation Details:** To align the meteorological data temporally, we treat each 6-hour period as one time step, averaging the pollutant concentrations within that period. During model training, we employ techniques such as regularization, gradient clipping, and progressive learning to improve training efficiency.

## E   Loss and Evaluation Metrics

Let $\mathbf{x} = (x_1, \ldots, x_m)$ represent the ground truth, and $\hat{\mathbf{x}} = (\hat{x}_1, \ldots, \hat{x}_m)$ represent the predicted pollutant concentrations.

The loss function used in this paper are defined as follows:

**Mean Squared Error (MSE) Loss**

$$MSELoss(\mathbf{x}, \hat{\mathbf{x}}) = \frac{1}{m} \sum_{i=1}^{m} (x_i - \hat{x}_i)^2 \tag{17}$$

The evaluation metrics used in this paper are defined as follows:

**Mean Absolute Error (MAE)**

$$\text{MAE}(\mathbf{x}, \hat{\mathbf{x}}) = \frac{1}{m} \sum_{i=1}^{m} |x_i - \hat{x}_i| \tag{18}$$

**Root Mean Square Error (RMSE)**

$$\text{RMSE}(\mathbf{x}, \hat{\mathbf{x}}) = \sqrt{\frac{1}{m} \sum_{i=1}^{m} (x_i - \hat{x}_i)^2} \tag{19}$$

## F  Ablation Study of Interpolation

As shown in 8, To investigate the effectiveness of the interpolation method used in our model, we conduct an ablation study comparing three variants: (1) our default RBF interpolation, (2) nearest-neighbor interpolation, and (3) a variant without any interpolation. As shown in Figure 8, the RBF interpolation achieves the best performance among the three settings, demonstrating its ability to provide smoother and more physically consistent spatial inputs, which benefits downstream prediction. Nearest-neighbor interpolation, while simple, introduces artifacts due to abrupt transitions between points. Removing interpolation altogether results in degraded performance, primarily because the model fails to effectively capture spatial continuity and interactions when operating on sparsely and irregularly distributed observational data.

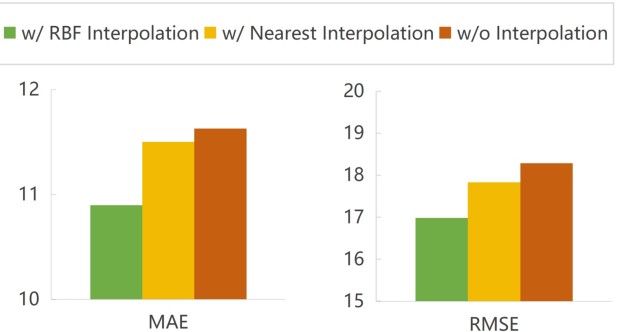

Figure 8: Ablation Study of Interpolation.

## G  Computational Complexity Analysis

The main computational cost of the proposed algorithm arises from the Eulerian ADR Decoder, including the computation of the three ADR terms and the three Predictor modules. The ADR terms consist of advection, diffusion, and reaction, each with a computational complexity of $\mathcal{O}(B \cdot H \cdot W)$. Each of the three Predictors (e.g., TAU3) contributes a computational complexity of $\mathcal{O}(B \cdot T \cdot D^2 \cdot H \cdot W)$, where $B$ is the batch size, $T$ is the length of the input sequence, and $D$ is the hidden dimension. Summing the complexities of all components, the total computational complexity is:

$$\mathcal{O}(B \cdot H \cdot W) + \mathcal{O}(B \cdot T \cdot D^2 \cdot H \cdot W) = \mathcal{O}(B \cdot T \cdot D^2 \cdot H \cdot W) \tag{20}$$

For reference, the computational complexity of graph-based methods is approximately $\mathcal{O}(B \cdot T \cdot D^2 \cdot N)$, where $N$ is the number of spatial nodes in the graph, which is typically smaller than $H \cdot W$, as shown in Table 3. On a single RTX 4090 GPU, the complete training process can be completed within 3 to 6 hours, depending on the data volume and forecast horizon. More importantly, during inference, generating a 72-hour forecast takes less than 0.15 seconds, which is fast enough to meet the requirements of most real-time applications.

Table 3: Comparison of Theoretical Computational Complexity between CTENet and graph-based Methods.

| Methods | CTENet | Graph-based Methods |
|---|---|---|
| Theoretical Complexity | $\mathcal{O}(B \cdot T \cdot D^2 \cdot H \cdot W)$ | $\mathcal{O}(B \cdot T \cdot D^2 \cdot N)$ |
| China Dataset | $\mathcal{O}(B \cdot T \cdot D^2 \cdot 80 \cdot 130)$ | $\mathcal{O}(B \cdot T \cdot D^2 \cdot 480)$ |
| USA Dataset | $\mathcal{O}(B \cdot T \cdot D^2 \cdot 70 \cdot 140)$ | $\mathcal{O}(B \cdot T \cdot D^2 \cdot 365)$ |

## H  Mathematical Analysis of the ADR Equation

The ADR equation 6 satisfies the existence, uniqueness, and stability conditions, making it a well-posed problem for modeling physical processes. A formal proof of these properties is provided below.

**Existence and Uniqueness:**  The ADR equation is a second-order linear parabolic PDE in both space and time. The well-known theory of existence for parabolic equations asserts that, for smooth (analytic) coefficients and appropriate initial and boundary conditions, there always exists at least one solution in a finite domain.

To prove the uniqueness of the solution for the ADR equation, we follow the standard method of analysis for parabolic partial differential equations. By using the energy method, we assume two solutions $X_1(t)$ and $X_2(t)$ satisfy the ADR equation. Defining the difference $w(t) = X_1(t) - X_2(t)$, we derive the following equation for $w$:

$$\frac{dw}{dt} + \vec{v} \cdot \nabla w = k_\theta \cdot \nabla^2 w + R'(X_1) \cdot w \tag{21}$$

Applying the energy method by multiplying both sides by $w$ and integrating over the spatial domain, we obtain:

$$\frac{d}{dt} \int_\Omega w^2 \, dx = -k_\theta \int_\Omega |\nabla w|^2 \, dx \tag{22}$$

This shows that $\int_\Omega w^2 \, dx$ is decreasing over time, implying that $w(t)$ tends to zero, which leads to the conclusion $X_1(t) = X_2(t)$. Hence, the ADR equation has a unique solution.

**Stability:**  For stability, we consider small perturbations in the initial condition. Let $X_0$ and $X_1$ be two initial conditions such that $|X_0 - X_1| < \epsilon$. The difference between the solutions $X_1(t)$ and $X_2(t)$ is given by:

$$w(t) = X_1(t) - X_2(t) \tag{23}$$

Using the energy method again, we find that the difference in the solutions remains bounded and decreases over time:

$$\int_\Omega w^2 \, dx \le \int_\Omega (X_0 - X_1)^2 \, dx \tag{24}$$

Thus, small changes in the initial condition lead to small changes in the solution, demonstrating the stability of the ADR equation. This confirms that the ADR equation is a well-posed problem under the given initial and boundary conditions.

## I  Boundary conditions

In the absence of external pollutant data beyond the study area, we apply Neumann boundary conditions [48] to maintain the stability and convergence of the numerical method. To minimize boundary effects, the computational domain is set at least 300 km away from the nearest monitoring station.

Neumann conditions enforce a zero-gradient (no-flux) condition at the boundaries of the domain:

$$\frac{\partial X}{\partial x} = 0, \quad \frac{\partial X}{\partial y} = 0 \quad \text{on the boundary} \tag{25}$$

## J   Statistical Significance Analysis

In this appendix, we derive an approximate probability that method A performs better than method B, based solely on their respective summary statistics: mean absolute error (MAE) and root mean square error (RMSE). Let these values be denoted as $\text{MAE}_A, \text{RMSE}_A$ for method A and $\text{MAE}_B, \text{RMSE}_B$ for method B.

Table 1 can be interpreted as paired tests between our method and each baseline model.

Let $y_i$ denote the ground truth value and $\hat{y}_i^{(A)}, \hat{y}_i^{(B)}$ denote the predictions of methods A and B on sample $i$. We define the absolute prediction error as:

$$d_i^{(A)} = \left| \hat{y}_i^{(A)} - y_i \right|, \quad d_i^{(B)} = \left| \hat{y}_i^{(B)} - y_i \right|. \tag{26}$$

We aim to estimate the probability that $d_i^{(A)} < d_i^{(B)}$, i.e., method A outperforms B on a randomly drawn sample. We assume the errors $d_i^{(A)}$ and $d_i^{(B)}$ are independent and approximately follow Gaussian distributions:

$$d_i^{(A)} \sim \mathcal{N}(\mu_A, \sigma_A^2), \quad d_i^{(B)} \sim \mathcal{N}(\mu_B, \sigma_B^2), \tag{27}$$

with parameters estimated from available statistics:

$$\mu_A \approx \text{MAE}_A, \quad \mu_B \approx \text{MAE}_B, \tag{28}$$

$$\sigma_A^2 \approx \text{RMSE}_A^2 - \text{MAE}_A^2, \quad \sigma_B^2 \approx \text{RMSE}_B^2 - \text{MAE}_B^2. \tag{29}$$

Define the difference in absolute errors:

$$\Delta d_i = d_i^{(B)} - d_i^{(A)}, \quad \Delta d_i \sim \mathcal{N}(\mu_B - \mu_A, \sigma_A^2 + \sigma_B^2). \tag{30}$$

Standardizing this with a normal variable:

$$Z = \frac{\Delta d_i - (\mu_B - \mu_A)}{\sqrt{\sigma_A^2 + \sigma_B^2}} \sim \mathcal{N}(0, 1), \tag{31}$$

The probability that method A outperforms method B is then given by:

$$p = P(d_i^{(A)} < d_i^{(B)}) = \Phi\left( \frac{\mu_B - \mu_A}{\sqrt{\sigma_A^2 + \sigma_B^2}} \right)$$

$$= \Phi\left( \frac{\text{MAE}_B - \text{MAE}_A}{\sqrt{(\text{RMSE}_A^2 - \text{MAE}_A^2) + (\text{RMSE}_B^2 - \text{MAE}_B^2)}} \right) \tag{32}$$

where $\Phi(\cdot)$ denotes the standard normal cumulative distribution function. As a result, the likelihood that method A performs better than method B on n samples is:

$$P(A \text{ better than } B) \approx 1 - \Phi\left( \frac{n/2 - np}{\sqrt{np(1-p)}} \right). \tag{33}$$

For all evaluations, each experiment includes at least 174 sequences, with each sequence containing observations from no fewer than 365 stations, as detailed in Appendix D. Each station provides a temporal sequence of observations, which is treated as an independent sample for statistical evaluation. Thus, the total number of samples is $n = 174 \times 365$. Based on our analysis, our algorithm outperforms the best SOTA baseline with statistical significance at a confidence level of at least 99.5%.

