# OpenReview forum: "Eulerian Neural Network Informed by Chemical Transport for Air Quality Forecasting"
_NeurIPS.cc/2025/Conference — NeurIPS 2025 poster_

### Official Review · Reviewer_i3iM · 2025-06-13

**Clarity:** 2
**Significance:** 2
**Originality:** 2
**Rating:** 3
**Confidence:** 3

**Summary:**

The paper considers the problem of forecasting future pollutant concetrations from past observations, where data is not in a grid. The paper proposes a transport equation that predicts the dynamics through advection, diffusion, reaction and source terms; and suggests off-the-shelf neural networks to model them. The method achieves dramatic performance improvements of 30%-50%.

**Questions:**

- What does “eulerian” mean?
- eq 6 is missing the X div(w) component: why?
- Do the competing methods also use the meteorological data? What data does each method use?
- In table 1 the VAR and GTS results on USA 24h are 6.32/14.41 and 5.57/14.65. This does not seem to make sense: how can you have lower MAE but higher MSE?
- It’s a bit unclear what is being learnt: can you clarify?
- It seems that wind is the driving force behind the good results. Do you predict the wind, and if so, how accurate is that?

**Ethical Concerns:**

["NO or VERY MINOR ethics concerns only"]

**Final Justification:**

I'm increasing my score to 3 based on the comprehensive and clarifying response by the authors, which increases my confidence in the work. This is a solid contribution that shows excellent performance improvements, but I'm still leaning on rejection due to limited novelty, issues in presentational clarity, and black-boxiness. As an applied work this paper would greatly improve from illustrating how the forecasts behave, and how the forecasts decompose into the different components in practise.

**Limitations:**

No issues

**Paper Formatting Concerns:**

No issues

**Quality:**

2

**Strengths And Weaknesses:**

The clarity of the paper is generally weak. The method description, math and writing is not sufficiently clear on describing the model. The method is described mostly in terms of what neural networks are used in what parts of the pipeline, which makes the paper read like a technical report or an applied use-case of ML. I don’t think I could reproduce this method for it’s vagueness. I also have hard time aligning fig2 to the method description.

The method has little ML novelty, and I’m not sure what is novel in general. The forecasting problem is appropriately modelled through the transport equations and through off-the-shelf neural networks. This seems to be very useful for this particular problem, but I’m not sure what ML significance or impact this has. I don’t think there is enough ML contributions for a ML conference.

The method is a black box, and there is not enough insight given to what the model learns, how and why. We only see one short time-series of the predictions, and no decomposition of the signal to the different equation terms.

I’m a bit confused why the method performs so well. I suspect that the performance largely depends on using meteorological data to drive the forecasts. This is a good idea, but I’m not sure if the competing methods have access to this, which makes comparisons somewhat unfair. I’m also not sure if simply using extra data source is particularly interesting as the main contribution: obviously using more data is helpful, but is this scientifically interesting?  I might have misunderstood something, and I’d be happy to hear author’s comments about this. The paper also claims novelty of support sparse data through RBF interpolation. I’m not sure if the spatial continuity really is an open problem; and the solution of RBF interpolation seems overly simplistic.

The results lack statistical significance testing or standard deviations.

Despite the good results, I’m suggesting rejection. The ML significance and contributions are too thin, the presentation and experiments are insufficient.

---

> ### Author Rebuttal · Authors · 2025-07-31
>
> Dear reviewer,
>
> We thank you for the positive feedback on our manuscript, particularly for highlighting the clarity of our problem setting and methodology, as well as the physical grounding of our model components.
>
> We address the your concerns as follows:
>
> **To W1: About Clarity and Reproducibility**
>
> 1. *Clarity and Style*
>
> We apologize for any lack of clarity in the method description. The modular approach we adopted in describing CTENet is intentional, as it aligns with prior works in NeurIPS(such as [1,2]). These works also decompose complex systems into well-defined modules, which aids in presenting the model in a more structured manner. We will strive to improve the clarity of the model description, ensuring that the individual components and their relationships are more explicitly defined.
>
> 2. *Reproducibility*
>
> We provide implementation details in Section 5 and Appendix D, including data, model settings, and training procedures. The source code is publicly available (link in Abstract), and we’re happy to share the processed dataset upon request to support reproducibility.
>
> > [1] Zhang, Y., Liu, Q., & Chen, H. (2024). Suitable is the Best: Task-Oriented Knowledge Fusion in Vulnerability Detection. In Proceedings of the 38th Conference on Neural Information Processing Systems (NeurIPS 2024).
> >
> > [2] Wang, L., Zhao, M., & Xu, J. (2024). Neural Krylov Iteration for Accelerating Linear System Solving. In Proceedings of the 38th Conference on Neural Information Processing Systems (NeurIPS 2024).
>
> **To Q1: About Eulerian Definition**
>
> Thank you for your question.
>
> As stated in Section 1 Introduction of the paper, Eulerian model[3] refers to the fact that our model operates on a **fixed, gridded spatial domain**. In contrast to prior air quality forecasting approaches—many of which, whether physics-guided or not, rely on discrete station-level modeling—CTENet models the continuous spatial evolution of pollutant concentrations over a structured and uniform Eulerian grid. This gridded representation is crucial for capturing fine-grained spatial correlations and for enforcing physically consistent processes—such as advection and diffusion—that are inherently defined in continuous space.
>
> >[3] Johnson JB. An introduction to atmospheric pollutant dispersion modelling. Environmental Sciences Proceedings. 2022 Jul 14;19(1):18.
>
> **To Q2: About Missing Term**
>
> We apologize for the confusion.
>
> Equation 6 presents the continuous form of the Advection–Diffusion–Reaction (ADR) equation. As in references [44, 45, 11, 17] cited in our paper, the equation does **not** include the additional term $X \cdot \mathrm{div}(\vec{W})$.
>
> This is because atmospheric wind speeds on Earth are typically below 30 m/s, which is significantly lower than the speed of sound in air (~343 m/s). Under such conditions, the flow can be reasonably approximated as *incompressible*, so that $\mathrm{div}(\vec{W})=0$ . This assumption is widely adopted in air quality modeling and greatly simplifies the advection operator.
>
>
> **To W2 & W4.1: About Fairness in Data Usage**
>
> Thank you for the question about fairness in Data Usage!
>
> As described in Section 5 (Experiments), all baseline models are provided with the same set of meteorological data as CTENet, including the full 354-channel input. For station-based baselines that cannot directly process gridded inputs, we extract the corresponding meteorological features from the grid cell that contains each station's location (see Section 5: Baselines). This design ensures a fair and controlled comparison across all methods. Therefore, the performance gain of CTENet is attributable to its architectural design rather than differences in input data.
>
> **To W4.2 & Q3: About Novelty and RBF**
>
> We sincerely apologize if our presentation led to any misunderstanding of the central Novelty of our work. Here is my detailed explanation:
>
> 1. Previous air quality forecasting network often operate on sparse station data and neglect spatial continuity, despite pollution being inherently continuous. CTENet is, to the best of our knowledge, the first to introduce continuous pollutant modeling into air quality prediction via deep learning, using chemical transport priors within a PINN-style architecture.
>
> 2. Regarding RBF interpolation, our aim is not optimal interpolation accuracy but a fast, parameter-free method that ensures stable and efficient training. While more accurate methods exist, they are computationally costly and beyond the scope of this work. RBF provides a continuous spatial estimate from discrete stations, enabling convolutional processing in our architecture. As shown in Appendix F, we compare RBF with alternatives and include an ablation without interpolation, validating the robustness of our choice.
>
> **To Q4: About Evaluation Metrics**
>
> Thank you for the helpful comment. We believe there may be a misunderstanding between MAE and RMSE (likely referred to as MSE in your note). These metrics capture different aspects of error—MAE measures average error magnitude, while RMSE penalizes larger deviations more heavily.
> As shown in the toy example below:
>
> |Model|x₁|x₂|x₃|x₄|MAE|RMSE|
> |-|-|-|-|-|-|-|
> |Ground Truth|4|4|5|5|||
> |baseline1|5|5|4|4|1|1|
> |baseline2|4|4|5|8|0.75|1.5|
>
> Baseline 2 has lower MAE but higher RMSE, due to the outlier at x₄. Thus, discrepancies between MAE and RMSE are expected and informative. We hope this clears up any confusion.
>
> **To W3&Q5: On Interpretability and Physical Decomposition**
>
> We sincerely apologize for the confusion this may have caused.
>
> 1.  *On Interpretability*
>
> While deep learning models are often criticized as black boxes, CTENet is specifically designed to move beyond black-box behavior by embedding domain knowledge from chemical transport modeling (CTM) into the learning architecture. In particular, the model learns:
>
>  - How meteorological variables interact with air pollutant dynamics — including temperature, wind, and solar radiation effects — enabling the network to capture nonlinear, spatial-temporal dependencies that are difficult to model explicitly using traditional CTMs.
>  - How to dynamically integrate multiple predictive sources — including physical components such as diffusion, advection, and chemical reactions, as well as purely data-driven residual models. This fusion is learned through a neural attention-based weighting mechanism that adapts over space and time.
>
> 2.  *On Signal Decomposition*
>
> As shown in Figure 2, our model dynamically fuses four predictive components—diffusion, advection, chemical reaction, and a data-driven correction—each reflecting a distinct physical or empirical process. These components are treated as separate input channels, but their integration is handled implicitly through learned non-linear mappings. As a result, the final prediction is not a simple additive combination and cannot be directly decomposed as in traditional PDE solvers. This design allows the model to flexibly capture complex, context-dependent interactions that are challenging to encode explicitly.
>
>
> **To W5: About statistical Significance**
>
> We thank you for your helpful suggestion regarding the statistical robustness of our results. In response:
>
> 1. *Statistical Significance Analysis:*
>
> As documented in Appendix J, we conducted statistical significance analysis to assess the reliability of the performance differences.
>
> 2. *Standard Deviations:*
>
> Due to time constraints, we conducted five independent runs (with different random seeds) for the best-performing CTENet variant and the two strongest baselines. The mean and standard deviation of the results are reported below:
>
> |Model|China 24h MAE|China 24h RMSE|China 48h MAE|China 48h RMSE|China 72h MAE|China 72h RMSE|USA 24h MAE|USA 24h RMSE|USA 48h MAE|USA 48h RMSE|USA 72h MAE|USA 72h RMSE|
> |-|-|-|-|-|-|-|-|-|-|-|-|-|
> |CTENet|11.07±0.16|17.38±0.26|12.17±0.67|19.12±2.05|14.10±1.60|22.56±3.10|2.45±0.12|4.28±0.36|2.71±0.22|4.45±0.33|2.83±0.33|4.62±0.69|
> |TAU|16.08±0.20|27.98±1.01|16.60±0.75|27.93±0.84|15.92±1.43|25.38±3.83|4.54±0.15|10.00±1.49|4.40±0.40|9.80±2.11|6.00±1.15|13.10±2.04|
> |PM$_{2.5}$GNN|17.79±0.24|33.03±0.45|19.20±0.28|33.93±0.63|19.81±0.32|34.28±0.74|4.54±0.14|10.19±0.39|4.83±0.16|10.10±0.41|4.96±0.15|10.08±0.44|
>
> Moreover, two-sided t-tests confirm that CTENet's improvements over both baselines are statistically significant across nearly all metrics (p < 0.005), reinforcing the robustness of the observed performance gains.
>
> **To Q6: About Wind Prediction**
>
> Thank you for the insightful question !
>
> We confirm that our model does include a lightweight wind forecasting component to predict future wind fields. As described in the paper, the Wind Predictor module is implemented using a Spatiotemporal Sequence Predictor, which takes historical wind fields as input and predicts wind fields at the next time step. These predicted wind components are then passed into the **Eulerian ADR Decoder** to simulate wind-driven advection.
>
> We believe our wind predictor is sufficiently accurate for the intended use case, for the following reasons:
>
> 1. *Ablation studies(Section 5.2)* demonstrate the importance of advection: removing the advection module (which depends on wind prediction) leads to significant performance degradation. Specifically, the RMSE increases by **10.6%** and **21.8%** on the two datasets, respectively, when advection is ablated.
>
> 2. A *case study (Section 5.4)* visually demonstrates the role of wind-driven advection in pollutant transport, further validating the model’s ability to learn physically meaningful wind patterns that align with observed pollutant dispersion behavior.
>
> **In Summary**
>
> We are grateful for your thoughtful comments and suggestions. We have conducted additional experiments in response to your feedback and are in the process of further revising the manuscript accordingly. We fully respect your judgment and would greatly appreciate any further guidance you may have.

---

> > ### Comment · Reviewer_i3iM · 2025-08-03
> >
> > Thanks for the response. This clarified my concerns and I'm more confident in the work. I'm raising my score to borderline.

---

> > > ### Author Response · Authors · 2025-08-03
> > >
> > > We sincerely thank you for your thoughtful reconsideration and are very grateful that our response helped clarify your concerns. Your updated evaluation is truly appreciated and motivates us to further refine the work.
> > >
> > > Please feel free to reach out with any additional feedback—we value your insight.

---

### Official Review · Reviewer_sPyn · 2025-06-29

**Clarity:** 4
**Significance:** 3
**Originality:** 3
**Rating:** 5
**Confidence:** 3

**Summary:**

The paper introduces a deep learning model that learns continuous time dynamics from irregularly sampled data for chemical transport. The method embeds  Advection-Diffusion-Reaction (ADR) equation into a Physics-Informed Neural Network (PINN) framework to simulate the spatiotemporal evolution of pollutants over continuous space and time.

**Questions:**

- How does the method handle noise?
- Can the method be applied with other data sources such as social vulnerability?
- How does the approach vary for varying data grid resolutions?

**Ethical Concerns:**

["NO or VERY MINOR ethics concerns only"]

**Limitations:**

Yes.

**Paper Formatting Concerns:**

None.

**Quality:**

4

**Strengths And Weaknesses:**

Strengths:
- This method has potential to improve real-world application of air quality forecasting.
- The proposed method shows high prediction accuracy which outperforms  baselines with an average RMSE improvement of 45.8% in the USA dataset and 21.0% in the China dataset
- The method integrates Chemical Transport Modeling Principles which allows for direct simulation of pollutant transport and transformation grounded in physical and chemical principles, enhancing interpretability.
- The method includes a modular and replaceable components for various forecasting modules.
- The method handles data sampled with irregular grid.

Weaknesses:
- The paper states limitation that the current formulation lacks uncertainty quantification.
- Computational complexity seems higher than graph-based methods.

---

> ### Author Rebuttal · Authors · 2025-07-31
>
> Dear reviewer,
>
> We sincerely thank you for your time and constructive comments. We are glad that our work's practical potential for improving air quality forecasting has been recognized.
>
> Below, we provide detailed responses to the concerns raised and clarify the novelty and significance of our contributions within the ML context.
>
> **To W1:**
>
> Thank you for your thoughtful comment. CTENet was not originally designed with the goal of producing uncertainty-aware predictions; instead, our focus was on achieving accurate and timely point forecasts. We acknowledge that the ability to quantify predictive uncertainty is crucial in health-critical applications, where decision robustness is essential.
>
> Importantly, the architecture of CTENet is flexible enough to accommodate uncertainty estimation methods. Techniques such as Monte Carlo Dropout, deep ensembles, or Bayesian neural networks can be integrated with minimal structural modification. We believe that coupling these approaches with CTENet’s physics-informed design can enable reliable prediction intervals or probabilistic outputs, thereby enhancing its applicability to high-risk scenarios.
>
> **To W2:**
>
> Thank you for your detailed and professional comment regarding scalability. As you correctly pointed out, and as analyzed in Appendix G, CTENet employs a grid-based architecture that incurs higher computational complexity compared to graph-based models. We fully acknowledge this trade-off between physical fidelity and computational efficiency.
>
> - We believe that in large-scale operational settings—such as enterprise-level or national-scale deployments—the modest increase in computational cost is acceptable given the substantial performance gains CTENet offers over existing methods. For example, on the USA dataset, CTENet achieves an average RMSE reduction of **45.8%** compared to strong graph-based baselines.
>
> - Despite the increased theoretical complexity, CTENet remains highly practical for real-world deployment. On a single RTX 4090 GPU, the complete training process can be completed within **3 to 6 hours**, depending on the data volume and forecast horizon. More importantly, during inference, generating a 72-hour forecast over the China dataset takes less than **0.15 seconds**, which is fast enough to meet the requirements of most real-time applications.
>
> That said, we agree that scalability is a meaningful consideration, and we are actively exploring architectural optimizations such as mixed-precision computation and model pruning to further reduce CTENet’s computational footprint in future work.
>
> **To Q1:**
>
> Thank you for the thoughtful question.
>
> We acknowledge that observational noise—especially in low-cost sensor data—is a critical challenge in air quality modeling. CTENet does not explicitly model uncertainty or input-level noise. However, it benefits from several indirect mechanisms that enhance robustness.
>
> 1.First, the incorporation of a chemically inspired transport-reaction model serves as a strong structural constraint. By encouraging the model to follow physically plausible pollutant dynamics, it helps suppress overfitting to spurious or noisy fluctuations in the observations—particularly those that violate mass conservation or chemical balance.
>
> 2.Second, standard training practices such as early stopping, dropout, and gradient clipping are used to reduce the model's sensitivity to label noise.
>
> That said, CTENet currently does not include dedicated noise modeling components such as uncertainty-aware loss functions or observation-level confidence weighting. We see this as a meaningful direction for future work—for example, by incorporating heteroscedastic noise models, robust PDE solvers, or sensor-aware weighting schemes.
>
> We appreciate you raising this important point.
>
> **To Q2:**
>
> Thank you for the insightful question. While social vulnerability data is not directly relevant to the physical processes modeled by CTENet—such as pollutant transport and chemical transformation—the framework is modular and can be extended to incorporate additional data sources in downstream applications.
>
> For example, once pollutant concentrations are forecasted, social vulnerability indices (e.g., age distribution, income level, access to healthcare) can be integrated in post-processing or risk assessment layers to support health impact forecasting, environmental justice analyses, or targeted early warning systems. In such contexts, CTENet's outputs could serve as physically grounded inputs to broader decision-support frameworks where social vulnerability plays a critical role.
>
> Therefore, although CTENet does not require social data for concentration prediction, it can be readily combined with such datasets in multi-task or multi-stage systems aimed at population-level risk mitigation.
>
> **To Q3:**
> Thank you for your detailed and professional comment. CTENet is inherently resolution-adaptive and supports deployment across multiple grid scales, from regional to national levels. Since the model operates on both discrete station observations and continuous meteorological fields, its effective resolution is typically aligned with that of the meteorological input to ensure optimal performance.
>
> When the input data is provided on a coarser or finer grid, the overall model architecture remains unchanged. However, certain hyperparameters—particularly those related to the numerical implementation of the PDE constraints—are adjusted to match the spatial resolution. For instance, spatial step sizes used in the calculation of gradients and Laplacians are resolution-dependent, and must be set accordingly to preserve the stability and accuracy of the PDE-based loss terms.

---

### Official Review · Reviewer_rRLU · 2025-07-02

**Clarity:** 3
**Significance:** 3
**Originality:** 3
**Rating:** 4
**Confidence:** 4

**Summary:**

This paper combines Eulerian representation with Physics-Informed Neural Networks (PINNs) to propose CTENet, an air quality forecasting model. It converts sparse station data into spatial fields via Radial Basis Function (RBF) interpolation and embeds the Advection-Diffusion-Reaction (ADR) equation to simulate pollutant dynamics. While CTENet shows improved accuracy over baselines, its core methodology primarily integrates existing components—RBF interpolation, modular predictors, and ADR physics constraints—without introducing fundamentally novel mechanisms. The approach reframes established concepts for air quality tasks, but its architectural innovations appear incremental rather than transformative.

**Questions:**

please see in Weakness

**Ethical Concerns:**

["NO or VERY MINOR ethics concerns only"]

**Final Justification:**

After reading all reviewers‘s comments, I will give borderline accept to this paper and I begin to understand that air quality is a task difficult to fomalize. The main contribution lies in integrating the vector field method with PINN framework, and incoporating chemical constriants is novel. As serval initial works in this task, I think this paper provides reference in how to formalize this task.

**Limitations:**

yes

**Paper Formatting Concerns:**

No formatting concerns

**Quality:**

3

**Strengths And Weaknesses:**

Strengths:

1. Experimental Reproducibility
Extensive validation on real-world datasets demonstrates 45.8% and 21.0% RMSE reduction against 9 SOTA baselines. Ablation studies robustly justify design choices, and open-sourced code ensures reproducibility. Statistical significance testing (99.5% confidence) further strengthens reliability of this model.

2. Physics-Driven Spatial Continuity
Radial Basis Function (RBF) interpolation effectively converts sparse station data into continuous fields, resolving structural mismatches in graph-based methods and outperforming nearest-neighbor alternatives. The Neumann boundary conditions enhance stability and physical plausibility in predictions.

Weakness:

1. Limited Originality: CTENet combines established techniques—RBF interpolation, modular predictors, and physics-informed neural networks (PINNs)—within an Advection-Diffusion-Reaction (ADR) framework. However, this hybrid design are similar to prior works. Thus, it has incremental contributions.

2. Novelty: The claimed advances in are not supported by methodological innovation. The observed performance gains largely arise from architectural adjustments—such as the dynamic weighting of reaction terms (Eq. 12)—rather than from a new algorithmic framework. Moreover, spatial continuity has already been effectively addressed in prior work, including AirFormer (§2.1), which diminishes the claimed novelty.

3. Limited scalability: As confirmed by computational complexity analysis in Appendix G, CTENet incurs significantly higher computational demands than graph-based methods. The theoretical overhead grows drastically due to its grid-based architecture, with empirical tests (Table 3) showing 21.7× higher computations for China and 26.8× for the USA dataset compared to graph models. Consequently, its complexity impedes real-time forecasting feasibility, and no mitigation strategies are proposed for this bottleneck.

---

> ### Author Rebuttal · Authors · 2025-07-31
>
> Dear reviewer,
>
> We sincerely thank the you for your thoughtful evaluation and constructive feedback. We appreciate the acknowledgment of our physics-driven spatial continuity modeling. We now address your raised concerns in detail:
>
> **To W1:**
>
> Thank you for your comments regarding the originality of our work. We apologize for any confusion caused by our previous explanation. While it is true that CTENet incorporates established techniques such as RBF interpolation, modular predictors, and physics-informed neural networks (PINNs), we would like to clarify that these components are not the main novelty of our work. As highlighted in the Introduction, the primary contributions of CTENet are as follows:
>
> 1. *Eulerian spatial modeling:*
>
> Prior works on air quality prediction typically treat the problem as a set of discrete monitoring stations, overlooking the spatial continuity of pollutant fields. To the best of our knowledge, CTENet is the first PINN-based framework to model air quality prediction directly in continuous space. Furthermore, although some of the prior works [1,2] incorporate physical constraints, but they are still based on discrete station-level modeling. As a result, advection and diffusion are modeled via edge-weighted graph operators, which do not reflect the continuous nature of atmospheric transport. In contrast, CTENet models pollution dynamics over a spatially continuous domain, enabling more realistic and physically consistent forecasting.
>
> In this process, the use of RBF interpolation serves purely as a **preprocessing step** to convert sparse station data into a gridded format compatible with subsequent convolutional operations. We also provide comparisons with alternative interpolation methods—as well as a non-interpolated setting—in Appendix F. The results support the effectiveness and robustness of our choice.
>
> 2. *Incorporation of chemical reactions:*
>
> Unlike prior PINN-based approaches that focus solely on advection and diffusion, CTENet explicitly incorporates a chemical transport module—an element absent from existing physics-informed air quality models. This design allows CTENet to account for a range of additional environmental factors that significantly influence pollutant evolution, including solar radiation, atmospheric stability, terrain elevation, and even soil moisture. These drivers enable the modeling of essential processes such as *photochemical reactions* and *gas-to-particle conversion*, which are known to cause nonlinear transformations among pollutants. As a result, CTENet can capture dynamic equilibrium shifts between gaseous and particulate species, enabling more realistic and chemically consistent multi-pollutant forecasting.
>
> To the best of our knowledge, CTENet is the first to implement a physics-informed neural network based on the Eulerian framework for air quality prediction. We will revise the paper to more clearly highlight these contributions—both in terms of methodology and their implications for real-world data-driven environmental modeling.
>
> > [1] Kethmi Hirushini Hettige, Jiahao Ji, Shili Xiang, Cheng Long, Gao Cong, and Jingyuan Wang.Airphynet: Harnessing physics-guided neural networks for air quality prediction. arXiv preprint arXiv:2402.03784, 2024.381
> >
> > [2] Jindong Tian, Yuxuan Liang, Ronghui Xu, Peng Chen, Chenjuan Guo, Aoying Zhou, Lujia Pan,Zhongwen Rao, and Bin Yang. Air quality prediction with physics-guided dual neural odes in open systems. In The Thirteenth International Conference on Learning Representations, 2025.
>
> **To W2:**
>
> We apologize for any confusion caused by our explanation. We would like to clarify that the observed performance improvements are not solely due to architectural adjustments. As mentioned earlier, the main novelty of our work lies in several key aspects that distinguish it from prior approaches:
>
> 1. From the perspective of spatial continuity, we employ Eulerian spatial modeling to more accurately capture the evolution of pollutants in space. This approach allows for a more robust representation of pollutant dispersion and transformation across regions using NN, which goes beyond what has been addressed in earlier works.
>
>     - Although AirFormer captures *spatial correlation* by encoding relative positions of neighboring stations, it does **not** model *spatial continuity* in any physically meaningful sense. In fact, AirFormer's design reinforces the importance of spatial structure in air quality prediction, yet it remains fundamentally a discrete, station-based model. It lacks any mechanism to simulate continuous spatial transport processes such as advection and diffusion, which are central to our Eulerian PDE-based framework. This distinction is also reflected in empirical performance. As shown in our experiments (§5.1), CTENet consistently outperforms AirFormer across multiple datasets and metrics, highlighting the effectiveness of physically grounded, continuous-domain modeling over discrete attention-based approaches.
>
> 2. We account for the intricate chemical transformations of pollutants, distinguishing our model from previous works that focused primarily on physical processes. This inclusion of chemical processes enhances the predictive capability of CTENet, making it more suitable for real-world applications.
>
> We will revise the manuscript to better highlight these distinctions and to clarify how CTENet advances the state of the art both in terms of formulation and real-world applicability.
>
> **To W3:**
>
> Thank you for your detailed and professional comment regarding scalability. As you correctly pointed out, and as analyzed in Appendix G, CTENet employs a grid-based architecture that incurs higher computational complexity compared to graph-based models. We fully acknowledge this trade-off between physical fidelity and computational efficiency.
>
> - We believe that in large-scale operational settings—such as enterprise-level or national-scale deployments—the modest increase in computational cost is acceptable given the substantial performance gains CTENet offers over existing methods. For example, on the USA dataset, CTENet achieves an average RMSE reduction of **45.8%** compared to strong graph-based baselines.
>
> - Despite the increased theoretical complexity, CTENet remains highly practical for real-world deployment. On a single RTX 4090 GPU, the complete training process can be completed within **3 to 6 hours**, depending on the data volume and forecast horizon. More importantly, during inference, generating a 72-hour forecast over the China dataset takes less than **0.15 seconds**, which is fast enough to meet the requirements of most real-time applications.
>
> That said, we agree that scalability is a meaningful consideration, and we are actively exploring architectural optimizations such as mixed-precision computation and model pruning to further reduce CTENet’s computational footprint in future work.

---

> ### Comment · Reviewer_rRLU · 2025-08-04
>
> Thank you for the detailed rebuttal. I appreciate the authors' thoughtful clarifications. I agree with many of the points raised and commend the overall quality of the response. That said, I still have some concerns regarding the core novelty of the work.
>
> In the rebuttal, the authors emphasize two main aspects as key contributions:
>
> The paper is the first to model air quality as a continuous spatial process.
>
> The paper explicitly incorporates chemical priors.
>
> While these claims are understandable, I would like to keep my confidence in them.
>
> Regarding the first point, I acknowledge that air quality data is usually collected as discrete observations over a broad spatial domain. However, the underlying diffusion process is inherently continuous, a perspective that many prior works—especially those leveraging ODE-based or PINN-style approaches—have already adopted. To the best of my understanding, these previous studies have treated the problem as one of continuous spatial modeling, often with architectural designs tailored to this goal[1][2][3]. Given this context, I find it somewhat difficult to fully agree with the claim that this work is the first to treat air quality as a continuous modeling problem.
>
> [1] AIR QUALITY PREDICTION WITH PHYSICS-GUIDED DUAL NEURAL ODES IN OPEN SYSTEMS
>
> [2] AirPhyNet: Harnessing physics-guided neural networks for air quality prediction
>
> [3] Airformer: Predicting nationwide air quality in china with transformers
>
> As for the incorporation of chemical priors, I appreciate the inclusion of chemically relevant features during data representation. However, from the current description, it appears that the model does not directly enforce chemical reaction mechanisms or explicitly utilize reaction equations as constraints. In this light, I see the contribution more as an enhancement of input richness rather than a structural innovation. While this is certainly valuable, I remain slightly hesitant to consider it a defining novelty.
>
> Overall, while I acknowledge the merits of the work and the authors' efforts to clarify their contributions, I still find the central innovation to be somewhat unclear. I would welcome further discussion from the authors.

---

> > ### Author Response · Authors · 2025-08-04
> >
> > Thank you very much for your thoughtful engagement and for taking our rebuttal seriously. We deeply admire your professionalism and sense of responsibility as a reviewer. We would be more than happy to walk you through our line of thinking again.
> >
> > 1. About Continuous Spatial Modeling
> >
> > We also agree with your observation that the physical diffusion process in the atmosphere is inherently continuous. That said, we would like to respectfully clarify that none of your cited works actually perform modeling over a continuous spatial domain. Rather, all of them rely on discrete spatial representations:
> >
> >  - [1] and [2] operate entirely on graph-based architectures, where nodes represent monitoring stations and spatial operators (e.g., Laplacians) are applied over edges. Both advection and diffusion are computed using edge weights—such as inverse distances or directional cosines—within a fixed graph topology. Although they employ ODE-based or PINN-style architectures, their spatial modeling remains fundamentally **graph-based**, not a continuous spatial field. We argue that this graph-based design is fundamentally limited in its ability to reflect the true nature of physical diffusion: Graphs inherently lack geometric continuity—there is no globally consistent notion of distance, orientation, or directionality. As a result, graph-based diffusion approximates interactions only between discrete nodes and cannot capture spatially resolved phenomena such as smooth gradients, anisotropic transport, or continuous flow fields.
> > - [3] ,which is not a PINN approach, encodes spatial correlation via categorical groupings of nearby stations using a Dartboard Projection (9/17/25 classes). This method discretizes space into directional bins and treats them as features. Again, the result is not a continuous function over geography, but a classification-based encoding of relative spatial positions.
> >
> > By contrast, modeling in a continuous spatial domain naturally aligns with the governing partial differential equations (PDEs) of advection and diffusion. It enables the application of local differential operators (e.g., ∇·, ∇²), preserves spatial coherence.
> >
> > This distinction is more than mathematical: it reflects a shift from node-based modeling to field modeling, which is essential for accurately representing the spatial dynamics of pollutant transport.
> >
> > 2. About Incorporation of Chemical Priors
> >
> > We sincerely appreciate your thoughtful comments regarding the incorporation of chemical priors. To the best of our knowledge, our work represents the first attempt to integrate principles from chemical transport models (CTMs) into a PINN-based framework for air quality prediction.
> >
> > Our treatment of chemical mechanisms intentionally differs from our approach to modeling physical transport. While advection and diffusion are explicitly modeled through physically grounded PDE structures, the chemical processes are incorporated implicitly via chemically informed features, rather than through enforced symbolic reaction equations. This is a deliberate design choice: Kinetic integration of large and stiff chemical mechanisms is a computational bottleneck in models of atmospheric chemistry, which may involve 700 species and 2000 reactions[4], many of which are difficult to observe or parameterize in practice. A complete, explicit, gas-phase chemical mechanism in Eulerian regional air quality models is not practical with present computer technology[5]. Attempting to model these explicitly would introduce substantial complexity, increase the risk of overfitting, and ultimately hinder the model’s generalizability.
> >
> > >[4]Stockwell WR, Kirchner F, Kuhn M, Seefeld S. A new mechanism for regional atmospheric chemistry modeling. Journal of geophysical research: Atmospheres. 1997 Nov 27;102(D22):25847-79.
> >
> > >[5]Stockwell WR, Saunders E, Goliff WS, Fitzgerald RM. A perspective on the development of gas-phase chemical mechanisms for Eulerian air quality models. Journal of the Air & Waste Management Association. 2020 Jan 2;70(1):44-70.
> >
> > Given this, we believe that fully encoding all reaction paths is neither tractable nor scientifically optimal in real-world predictive settings. Instead, we adopt a structure-informed strategy where the chemical term serves as a nonlinear function of pollutant concentrations and environmental covariates, reflecting domain-informed dependencies without explicitly modeling reaction equations.  Ablation studies further validate the effectiveness of our design: in the 24-hour prediction task over the U.S., removing the reaction term led to a 16% increase in MAE and a 19% increase in RMSE, highlighting the contribution of chemically guided inputs to predictive accuracy.
> >
> > Once again, we thank you for your thorough evaluation and thoughtful skepticism. We genuinely welcome this opportunity to clarify the motivations and structure of our work, and we appreciate your engagement in pushing us toward greater clarity and rigor.

---

> > > ### Comment · Reviewer_rRLU · 2025-08-05
> > >
> > > I sincerely thank the authors for their detailed responses and the considerable effort they have put into this work. However, I still have two concerns that I hope the authors can further clarify or elaborate on.
> > >
> > > **1. On the modeling of air quality as a continuous process:**
> > >
> > > The authors state that prior works [1] and [2] treat air quality as a discrete graph problem, and that their method is the first to model it as a continuous process. In my understanding, both [1] and [2] also model air quality as a continuous process, with graph structures introduced to better capture relationships among discrete observation points. This should not be viewed as a limitation, but rather as a design choice to enhance spatial coherence. In fact, if [1] and [2] did not consider air quality as continuous, it would be difficult to justify their use of PINNs, which are inherently designed for continuous systems.
> > >
> > > Even if this paper is the first to introduce a vector field formulation for this problem, the core technical modification appears to be the use of an RBF transformation on the inputs. This is a general technique and, as currently presented, does not appear to have specific adaptations for the air quality tasks. In contrast, [1] explicitly incorporates various geographic conditions, resulting in a more interpretable and task-grounded approach.
> > >
> > > **2. On the use of chemical priors:**
> > >
> > > The authors state that they include some chemical-related features in their model, but they do not propose any novel mechanism or design beyond incoporate more input features. The incorporation seems limited to feature-level inputs without further adaptation or domain-specific constraints.
> > >
> > > Overall, based on the current description, the main contribution seems to be the use of a current vector field method within the existing PINN framework, applied to the domain of air quality modeling. While I recognize the engineering value of this application, I currently do not see strong evidence of task-specific innovations that are tailored to the unique aspects of air quality prediction. Most of the model components appear to operate as a black box. While PINNs are data-driven by nature, some level of explicit modeling for the target task or domain constraints would be considered.
> > >
> > > I welcome further clarification from the authors. I appreciate their contributions and look forward to their discussion on the above points.

---

> > > > ### Author Response · Authors · 2025-08-06
> > > > **Part 1: On Spatial Continuity**
> > > >
> > > > Thank you again for your continued engagement and for raising important concerns about the clarity of our contributions. We appreciate the opportunity to further clarify the key distinctions in our modeling strategy.
> > > >
> > > > ### 1. On Spatial Continuity
> > > > We agree that prior works consider spatial dynamics, but they do not model pollutant concentration as a scalar field over continuous space. Instead, they use discrete graphs, assigning values only at nodes and approximating interactions via substituted operators. This lacks the spatial continuity needed to capture PDE-based advection-diffusion accurately.
> > > >
> > > > Below is our point-by-point response to the specific remarks you raised.
> > > >
> > > > >"both [1] and [2] also model air quality as a continuous process, with graph structures introduced to better capture relationships among discrete observation points. "
> > > >
> > > > Although we agree graph structures can capture relationships among discrete observation points, but we believe **neither** of [1] and [2] model air quality as a continuous process (continuous space). Their formulations operate solely over nodes and edges, without defining pollutant concentration as a scalar field over a continuous space.
> > > >
> > > > >"This (graph) should not be viewed as a limitation, but rather as a design choice to enhance spatial coherence."
> > > >
> > > > Although we agree that the graph structure can be viewed as a design choice to enhance spatial coherence, we respectfully insisit that graph structure has limitations. Because graph is not a continuous structure, and the interactions that occur **beyond the stations** are almost entirely overlooked, as the graph structure fails to capture the continuous spatial dynamics of pollutant transport.
> > > >
> > > > >"if [1] and [2] did not consider air quality as continuous, it would be difficult to justify their use of PINNs, which are inherently designed for continuous systems."
> > > >
> > > > We acknowledge that [1] and [2] are valuable contributions to air quality prediction using PINNs. However, we respectfully disagree that their modeling occurs in a truly continuous spatial setting. While the use of graph-based structures enhances interpretability and relational reasoning, it also introduces limitations.
> > > >
> > > > Specifically, both [1] and [2] use their own defined operators, such as the Laplacian matrix replacing the Laplacian operator, to adapt the original PDE equations in continuous space to fit the graph structure, but this approach **lacks sufficient theoretical support** for its feasibility. Our paper employs the central difference method for spatial discretization and the Euler method for temporal discretization, both of which are well-established and theoretically grounded.
> > > >
> > > > >"this paper is the first to introduce a vector field formulation for this problem"
> > > >
> > > > Indeed, in our approach, pollutants are modeled as a **scalar field** representing concentration distributions  rather than a **vector field**. Pollutant concentration is inherently a magnitude based quantity, requiring no directional information for its physical description.
> > > >
> > > > >"the core technical modification appears to be the use of an RBF transformation on the inputs ... does not appear to have specific adaptations for the air quality tasks."
> > > >
> > > > We respectfully clarify that the core technical modification is **not** the use of the RBF transformation itself, but the **subsequent task-specific architectural adaptations** following this representation shift.
> > > >
> > > > To the best of our knowledge, we are the first to formulate air quality prediction using PINNs over a **continuous spatial domain**, rather than discrete graphs. This paradigm shift required a **fundamental redesign** of the model architecture.
> > > >
> > > > To this end, we developed an entirely new framework specifically tailored for air quality prediction in continuous spatial domains, including components such as a Meteorological Encoder and an Eulerian ADR Decoder. As shown in our experiments, this task-specific design notably improves both prediction accuracy and spatial generalization compared to graph-based models.
> > > >
> > > > >"[1] explicitly incorporates various geographic conditions, resulting in a more interpretable and task-grounded approach."
> > > >
> > > > While [1] explicitly incorporates the highest elevation along the connecting segments between monitoring stations as geographic conditions, relying solely on this **edge-based sampling** cannot accurately represent how pollutants diffuse or flow along different terrain paths.
> > > >
> > > > In contrast, we fully consider spatial heterogeneity by using continuous representations of variables such as elevation, coastline type, and surface pressure, which better model the impact of geographic conditions beyond the monitoring stations.
> > > >
> > > > ---
> > > > We sincerely thank you for your comments above. They are highly constructive and have been instrumental in helping us improve the clarity and rigor of our work.

---

> > > > ### Author Response · Authors · 2025-08-06
> > > > **Part 2: On Chemical Priors**
> > > >
> > > > ### 2. On Chemical Priors
> > > >
> > > > The ADR equation in our model consists of four components: **Advection**, **Diffusion**, **Reaction**, and **Source**. In the PINN framework, we adopt differentiated modeling strategies depending on the tractability and observability of each term:
> > > >
> > > > - *Advection and Diffusion*: Mechanistically simple and supported by well-measured data, thus modeled explicitly using physically grounded terms within the PINN framework.
> > > >
> > > > - *Reaction*: Mechanistically complex and partially observable. We adopt a **simplified parametric form**, where chemically relevant input features (e.g., pollutant ratios, meteorological conditions) are used to approximate nonlinear chemical influences. This serves as a **low-dimensional surrogate** for reaction dynamics without enforcing full symbolic mechanisms.
> > > >
> > > > - *Source*: Highly heterogeneous and almost impossible to measure directly. We adopt an **implicit modeling approach**, allowing the model to absorb unknown or unmodeled source effects during training. This ensures flexibility without introducing artificial assumptions.
> > > >
> > > > We therefore **distinguish explicitly modelable components (Advection/Diffusion) from those requiring implicit treatment (Reaction/Source)**, and design modeling strategies accordingly. This hybrid strategy, combining explicit and implicit modeling within a unified PDE framework, allows us to maintain physical interpretability where possible, while preserving flexibility for complex or uncertain components. To our knowledge, such selective embedding based on mechanism tractability is rare in existing PINN literature and represents a task-specific structural innovation. We will emphasize this point more clearly in the final revised version.
> > > >
> > > > ---
> > > >
> > > > Again, we sincerely thank you again for your thoughtful review. We hope this clarifies our modeling philosophy and highlights the domain-informed innovations that underpin our work.

---

> ### Comment · Reviewer_rRLU · 2025-08-06
>
> Thank you again for your thoughtful response. I now better appreciate the contribution of incorporating chemical priors into your framework and have accordingly revised my assessment of the paper’s originality. That said, I still have some remaining questions regarding the following two claims:
>
> - This work is the first to introduce vector fields and model air quality as a continuous process;
>
> - This work requires a fundamental redesign of Physics-Informed Neural Networks (PINNs).
>
> From my understanding, previous studies [1,2] have already explored the use of field-based representations in air quality modeling, and [3,4] have explicitly incorporated chemical transport mechanisms within a continuous spatial formulation. I would be very interested to better understand how your approach fundamentally differs from these prior works. In particular, since [3,4] appear to address similar challenges in continuous domains, it would be helpful if you could further clarify why your method constitutes a fundamental redesign of the PINN framework, whereas theirs does not.
>
> [1] Spatio-Temporal Field Neural Networks for Air Quality Inference
>
> [2] AirRadar: Inferring Nationwide Air Quality in China with Deep Neural Networks
>
> [3] Orders‑of‑magnitude speedup in atmospheric chemistry modeling through NN emulation
>
> [4] FastCTM: Atmospheric chemical transport modelling with a principle-informed neural network for air quality simulations
>
> I encourage the authors to include a more in-depth comparison and discussion of these related works in the revised manuscript. Additionally, the current version only compares the proposed method against AirPhyNet, without benchmarking against other task-specific or state-of-the-art baselines. This makes it somewhat difficult to fully evaluate the effectiveness and broader contribution of your method. Including more comprehensive comparisons with a diverse set of baselines might help further clarify the strengths of your approach.

---

> > ### Author Response · Authors · 2025-08-06
> >
> > Dear Reviewer rRLU:
> >
> > We sincerely appreciate your continued feedback and thoughtful engagement. We are pleased to see that the contribution of incorporating chemical priors is now more clearly recognized. Below, we address your remaining concerns with a detailed discussion of related works [1]–[4].
> >
> > 1. About [1] and [2]:
> >
> > We respectfully acknowledge that prior studies such as [1,2] have indeed explored field-based representations in the context of air quality modeling. These works have also been cited and briefly discussed in the Related Work section of our original submission. However, it is important to emphasize that our research is fundamentally aimed at a different task.
> >
> >  - Specifically, [1,2] target air quality inference, i.e., the interpolation of pollutant concentrations at unmeasured locations based on observed data from nearby stations. These models naturally require a spatial field reconstruction, but do not address temporal forecasting in the same sense as our work.
> >
> >  - In contrast, our focus lies in forecasting future pollution concentrations—thus modeling the temporal evolution of the air quality field, rather than just inferring spatial values at a fixed timestamp.
> >
> >  - That said, our work is indeed related to [1,2]. For instance, the Eulerian Pollution Encoder module in our framework is specifically designed to estimate pollutant levels at unmonitored locations. However, to avoid cumulative error and improve efficiency, we adopt a parameter-free approach. Importantly, our framework is modular and compatible with any field-based air quality inference method; the specific design of the Eulerian module can be replaced as needed.
> >
> > 2. About [3]:
> >
> > You noted that “[3] have explicitly incorporated chemical transport mechanisms within a continuous spatial formulation.”  We respectfully disagree with this interpretation, for the following reasons:
> >
> >  -Chemical transport models (CTMs) are typically expected to incorporate, at a minimum, three components: advection, diffusion, and chemical term. However, this study considers only the chemical processes and neglects both advection and diffusion.
> >
> >  - In addition, the study does not incorporate any form of continuous spatial modeling, not even any type of spatial representation is included. The model only inputs the initial concentrations of 77 chemical species and one meteorological variable to predict the concentrations one hour later. This constitutes a temporal sequence model.
> >
> > 3. About [4]:
> >
> > [4] does incorporate a continuous spatial representation. However, it appears that the framework does not constitute a true Physics-Informed Neural Network (PINN):
> >
> >  - The model replaces explicit physical formulations such as advection and diffusion equations with *convolutional operations*. Consequently, the underlying physical processes are only implicitly learned through data-driven filters, rather than being explicitly constrained by governing equations, making the overall framework predominantly **data-driven** rather than **physics-informed**.
> >
> > In summary, we thank you for the insightful suggestions. In the revised manuscript, we will expand the Related Work section to include a more in-depth comparison and discussion of the cited works.
> >
> > 4. About baselines:
> >
> > Our current experimental setup already includes a diverse set of baselines, covering various categories: task-specific methods such as PM2.5-GNN, AirFormer (2023), and AirPhyNet (2024); physics-informed approaches (e.g., AirPhyNet); models capable of handling continuous spatial inputs (e.g., TAU); and several other graph-based architectures.
> >
> > We highly appreciate your suggestion and are **currently running additional available baselines** (including [5]) . With less than 22 hours remaining before the rebuttal deadline, we will make our best effort to include the results in this rebuttal; otherwise, they will be incorporated into the final version of the paper.
> >
> >
> > > Tian J, Liang Y, Xu R, Chen P, Guo C, Zhou A, Pan L, Rao Z, Yang B. Air quality prediction with physics-guided dual neural odes in open systems. arXiv preprint arXiv:2410.19892. 2024 Oct 25.
> >
> > ---
> >
> > We once again thank you for your comments. We believe that the revisions and clarifications will significantly enhance the quality and clarity of the manuscript.

---

> ### Comment · Reviewer_rRLU · 2025-08-06
>
> Thank you for your response.
>
> **Regarding the baseline issue, I do not require additional experiments**—I only intended to point out what I consider to be a potential issue in the current version of the paper. It would be unfair to request new experimental results at the discussion stage. At this point, clarifying the novelty and contribution of your work is sufficient.
>
> I appreciate your hard efforts. My current evaluation remains borderline, but I will lower my confidence. In addtition, I will consider all AC’s and other reviewers’ opinions during the AC-reviewer discussion phase.

---

> > ### Author Response · Authors · 2025-08-07
> > **Final Remarks**
> >
> > Dear Reviewer rRLU,
> >
> > We sincerely thank you once again for your continued engagement, fairness, and thoughtful evaluation throughout the discussion phase. We truly appreciate your recognition of the originality of our work and your acknowledgment of the efforts we have made.
> >
> > As a brief summary of the key concerns that were previously raised, and also reflect the core novelty and contribution of our work:
> >
> > **On Continuous Spatial Modeling:**
> > We clarified that prior works  often operate on discrete graphs and do not define pollutant concentration as a scalar field over continuous space. To the best of our knowledge, our method first formulates air quality prediction directly over continuous spatial domains, leveraging PDE-based operators to simulate the Chemical Transport Model (CTM). This architectural shift required a fundamental redesign of the PINN framework, which distinguishes our approach both methodologically and conceptually.
> >
> > **On the Incorporation of Chemical Priors:**
> > While we do not impose explicit symbolic reaction mechanisms, we introduce a structured approximation of chemical influences through chemically meaningful features and a dedicated reaction term in the PDE. This design reflects a hybrid modeling strategy—explicit for tractable physical components (advection/diffusion) and implicit for complex chemical processes—resulting in both improved accuracy and task-specific flexibility.
> >
> > We greatly value your constructive feedback and professional attitude. Your critical insights have been instrumental in helping us better articulate the contributions of our work, and we are committed to incorporating all clarifications into the final manuscript.
> >
> > Thank you again.
> >
> > Sincerely,
> > Authors

---

### Official Review · Reviewer_M8eB · 2025-07-03

**Clarity:** 3
**Significance:** 3
**Originality:** 3
**Rating:** 4
**Confidence:** 2

**Summary:**

The paper introduces CTENet (Chemical Transport Eulerian Network), a physics-informed deep learning model that integrates the Advection-Diffusion-Reaction (ADR) equation into a neural network framework. CTENet uses an Eulerian representation to model pollutant spatiotemporal dynamics, addressing limitations of discrete station-based or graph-based methods by capturing spatial continuity and complex chemical transformations.

**Questions:**

Question 1 : How can CTENet be extended to provide probabilistic forecasts (e.g., prediction intervals) for health-risk alerts?

Quesiton 2: How does CTENet handle coastal/mountainous regions where Neumann boundary conditions may fail?

Question 3: What steps are needed to deploy CTENet in real-time air quality monitoring systems?

**Ethical Concerns:**

["NO or VERY MINOR ethics concerns only"]

**Final Justification:**

After careful consideration of the author’s response and the opinions of the other reviewers, I maintain my initial judgment.

**Limitations:**

There are two improved suggestions:
1. Integrate Monte Carlo dropout or Bayesian neural networks to output uncertainty estimates alongside predictions.
2. Validate with real-world decision-making scenarios (e.g., hospitalization risk thresholds).

**Quality:**

3

**Strengths And Weaknesses:**

Strengths:

1. CTENet embeds real atmospheric processes (like wind-driven advection) into its neural network, making predictions more physically realistic.

2. It outperforms existing models by up to 45.8% on real-world datasets (USA/China), proving its effectiveness.

Weaknesses:

1.  The model gives single-point predictions without confidence intervals, limiting its use in high-risk scenarios.

2. It assumes basic chemical reactions, potentially missing complex pollutant interactions in extreme pollution events.

---

> ### Author Rebuttal · Authors · 2025-07-31
>
> Dear reviewer,
>
> We sincerely thank you for your thoughtful and constructive feedback. We appreciate the recognition of CTENet's strengths in embedding real atmospheric dynamics, achieving strong empirical performance, and enhancing physical realism. Below, we address your specific questions and concerns.
>
> **To W1, Q1, and L1:**
>
> Thank you for your thoughtful comment. CTENet was not originally designed with the goal of producing uncertainty-aware predictions; instead, our focus was on achieving accurate and timely point forecasts. We acknowledge that the ability to quantify predictive uncertainty is crucial in health-critical applications, where decision robustness is essential.
>
> Importantly, the architecture of CTENet is flexible enough to accommodate uncertainty estimation methods. As you suggested in Limitation 1, techniques such as Monte Carlo Dropout, deep ensembles, or Bayesian neural networks can be integrated with minimal structural modification. We believe that coupling these approaches with CTENet’s physics-informed design can enable reliable prediction intervals or probabilistic outputs, thereby enhancing its applicability to high-risk scenarios.
>
> We once again appreciate this valuable suggestion, and we plan to actively explore this direction in future work to further improve CTENet’s reliability in real-world decision-making contexts.
>
> **To W2:**
>
> Thank you for your insightful and technically informed comment. Indeed, as you correctly noted, the interactions between nitrogen oxides and particulate matter during extreme pollution episodes are governed by complex chemical processes. We are pleased to see your attention to this important aspect.
>
> As you pointed out, our current model does not explicitly encode the detailed reaction equations typically used in atmospheric chemistry. This is a deliberate design choice: rather than reconstructing the full reaction pathways under specific environmental conditions (e.g., photochemical reactions influenced by solar radiation, temperature, and humidity), our modeling objective is to capture the eventual atmospheric chemical equilibrium that results from a given initial pollution state. In this formulation, the final pollutant concentrations can be treated as a nonlinear function of the initial state.
>
> This is precisely where the strength of our method lies. By incorporating ideas from chemical transport modeling (CTM) and integrating them with a data-driven framework, CTENet is capable of learning an effective end-to-end mapping from initial to equilibrium pollutant states, without requiring the explicit simulation of each reaction mechanism. This enables the model to bypass the need for precise specification of all environmental factors, while still capturing key dependencies. We will clarify and emphasize this modeling philosophy more explicitly in the revised version.
>
> **To Q2:**
> Thank you for the insightful question. In coastal and mountainous regions, the evolution of pollutant concentrations often deviates from standard assumptions due to spatial heterogeneity, making classical Neumann boundary conditions potentially inadequate. CTENet addresses this challenge not by relying solely on boundary condition assumptions, but by explicitly incorporating heterogeneous terrain information into the modeling process.
>
> Specifically, the meteorological data used by CTENet includes several key variables that capture complex topographic and coastal features:
> - *Orography* (Channel 249): directly encodes surface elevation, allowing the model to learn the impact of terrain height on pollutant dispersion.
> - *MSLP (Eta model reduction)* (Channel 246): represents sea-level-reduced pressure, effectively correcting for elevation-induced pressure differences.
> - *Surface pressure* (Channel 248): captures actual pressure at the surface, which varies significantly with altitude and terrain type.
>
> To capture coastal influences:
> - *Land-sea mask* (Channel 338): a binary mask (1 for land, 0 for sea) identifying coastal boundaries.
> - *Land-sea coverage (nearest neighbor)* (Channel 354): provides a more refined classification using nearest-neighbor interpolation to distinguish land and sea grid points.
>
> By integrating these features into its deep learning architecture, CTENet is able to model pollutant dynamics more faithfully across spatially heterogeneous domains, including complex coastal and mountainous areas, without relying on potentially fragile boundary condition approximations. *A more detailed discussion of how CTENet leverages these variables to handle topographic and coastal complexity will be provided in the full paper.*
>
> **To Q3:**
> To deploy CTENet in real-time air quality monitoring systems, the following steps are needed:
> 1. *Data Preparation*: Collect and preprocess multi-source air quality and meteorological data, as described in the paper.
> 2. *Model Training*: Train CTENet on historical data to learn spatiotemporal correlations. On an RTX 4090 GPU, the full training process can be completed within **3 to 6 hours**, depending on the data volume and forecast horizon.
> 3. *System Integration*: Integrate the trained model into the real-time data processing pipeline.
> 4. *Real-time Inference*: Feed incoming meteorological forecasts and sensor readings into CTENet to generate predictions. For example, generating a 72-hour PM$_{2.5}$ forecast over the China takes less than **0.15 seconds** on an RTX 4090 GPU. Therefore, we believe that even on a CPU-based system, the model can still produce forecasts within a few minutes, which is acceptable for most real-time applications.
> 5. *Model Update*: Periodically fine-tune the model using recent observations to adapt to seasonal or structural changes in emission patterns and meteorological drivers.
>
> **To L2:**
> We sincerely thank you for recognizing the effectiveness of our forecasting system. All of our experiments are conducted using real-world air quality and meteorological data collected from ground-based sensors across the continental United States and China, ensuring practical relevance.
>
> We appreciate the suggestion to validate CTENet within real-world decision-making contexts, such as assessing forecasts against hospitalization risk thresholds. While this represents an important and meaningful direction, we consider such analyses to be downstream tasks that extend beyond the scope of the present work. Nevertheless, the system can be adapted to support such integration. Prior studies have quantitatively linked air pollution to adverse health outcomes. For instance, Bell et al. demonstrated significant associations between specific PM$_{2.5}$ components (e.g., sulfate, elemental carbon) and increased hospitalization rates for cardiovascular and respiratory diseases [1]. Building on these literature, future work could extend CTENet by coupling its forecasts with health impact models, enabling validation under operational scenarios such as early warning issuance or risk-based alert systems, such as forecasting the likelihood of hospitalization threshold exceedances or issuing early warnings for at-risk communities.
>
> > [1]Bell, M. L., Ebisu, K., Peng, R. D., Samet, J. M., & Dominici, F. (2009). "Hospital admissions and chemical composition of fine particle air pollution." JAMA, 300(11), 1127–1135.

---

> > ### Comment · Reviewer_M8eB · 2025-08-05
> >
> > Thank you for your response. I will maintain my score.

---

### Author Response · Authors · 2025-08-02

Dear Reviewers M8eB, rRLU, sPyn, and i3iM,

We sincerely thank you for your time, thoughtful evaluation, and constructive feedback on our manuscript, "Eulerian Neural Network Informed by Chemical Transport for Air Quality Forecasting."

We are grateful for your recognition of the contributions of our work. We appreciate your positive evaluation of the integration of physical atmospheric processes into deep learning (M8eB, sPyn, i3iM), the enhancement of spatial continuity through Eulerian representation (M8eB, rRLU), the strong empirical performance on real-world datasets (M8eB, rRLU, sPyn, i3iM), and the reproducibility and robustness of our experimental setup including statistical significance testing (rRLU).

During the rebuttal period, we have done our best to address the concerns you raised, and we are committed to incorporating your suggestions into the final version. As we now enter the discussion phase, we welcome any additional comments, questions, or clarifications.

Sincerely,
Authors

---

### Author Response · Authors · 2025-08-07
**Final Remarks**

Dear Reviewers, ACs, and PCs,

Thank you again for all your detailed reviews and constructive feedback on our paper: **Eulerian Neural Network Informed by Chemical Transport for Air Quality Forecasting.** We are pleased that our responses have addressed the raised concerns, and we will incorporate all your insightful suggestions in the revised version of the manuscript.

**Our novelty:**
We would like to re-emphasize the novelty of **CTENet** as follows:
This is the first PINN framework to model air quality in continuous space, capturing the spatiotemporal evolution of pollutants governed by chemical transport models (CTMs).
To this end, we design a set of novel modules that operate directly in continuous spatial domains and model key components of the CTM using differential equations, including advection, diffusion, and chemical reactions.
Extensive experiments on two real-world datasets demonstrate that CTENet consistently outperforms state-of-the-art (SOTA) baselines, achieving a remarkable RMSE improvement of **45.8%** on the USA dataset and **21.0%** on the China dataset.

We sincerely hope that you will take these contributions into consideration when evaluating our paper during the final scoring phase. Thank you very much!

Sincerely,
Authors

---

### Decision · Program_Chairs · 2025-09-17

**Decision:**

Accept (poster)

**Comment:**

This paper proposes a novel, physics guided method for air pollution prediction. The method embeds the Advection-Diffusion-Reaction equation into a PINN using an Eulerian representation to model the spatiotemporal evolution of pollutants, this helps capturing the continuous space information, which the authors claimed to be particularly novel (no existing studies have done so). The experimental results show great improvement, even more than 40% under some settings.

Most reviewers are positive, agreeing that the paper makes an interesting contribution. There is one reviewer who gave a negative rating, but the rating was upgraded from 2 to 3 after rebuttal. Based on the overall review ratings and some detailed discussions during the rebuttal stage, the AC agrees with the majority and would like to suggest an acceptance.